# The flow responsive transcription factor Klf2 is required for myocardial wall integrity by modulating Fgf signaling

Seyed Javad Rasouli[1], Mohamed El-Brolosy[1], Ayele Taddese Tsedeke[1], Anabela Bensimon-Brito[1], Parisa Ghanbari[2], Hans-Martin Maischein[1], Carsten Kuenne[3], Didier Y Stainier[1]*

[1]Department of Developmental Genetics, Max Planck Institute for Heart and Lung Research, Bad Nauheim, Germany; [2]Department of Cardiac Development and Remodeling, Max Planck Institute for Heart and Lung Research, Bad Nauheim, Germany; [3]Bioinformatics Core Unit, Max Planck Institute for Heart and Lung Research, Bad Nauheim, Germany

**Abstract** Complex interplay between cardiac tissues is crucial for their integrity. The flow responsive transcription factor KLF2, which is expressed in the endocardium, is vital for cardiovascular development but its exact role remains to be defined. To this end, we mutated both *klf2* paralogues in zebrafish, and while single mutants exhibit no obvious phenotype, double mutants display a novel phenotype of cardiomyocyte extrusion towards the abluminal side. This extrusion requires cardiac contractility and correlates with the mislocalization of N-cadherin from the lateral to the apical side of cardiomyocytes. Transgenic rescue data show that *klf2* expression in endothelium, but not myocardium, prevents this cardiomyocyte extrusion phenotype. Transcriptome analysis of *klf2* mutant hearts reveals that Fgf signaling is affected, and accordingly, we find that inhibition of Fgf signaling in wild-type animals can lead to abluminal cardiomyocyte extrusion. These studies provide new insights into how Klf2 regulates cardiovascular development and specifically myocardial wall integrity.

DOI: https://doi.org/10.7554/eLife.38889.001

*For correspondence:
Didier.Stainier@mpi-bn.mpg.de

## Introduction

The embryonic heart must undergo a series of complex cellular and molecular processes to form a mature pumping organ (*Auman and Yelon, 2004*; *Bakkers et al., 2009*; *Staudt and Stainier, 2012*; *Milgrom-Hoffman et al., 2014*; *Dietrich et al., 2014*; *Staudt et al., 2014*; *Wilsbacher and McNally, 2016*; *Del Monte-Nieto et al., 2018*; *Miao et al., 2018*). Defects in any of these processes can lead to congenital heart malformations, which have estimated occurrences ranging from 4 to 50 cases per 1000 live births, and represent the most prevalent category of congenital anomalies (*Hoffman and Kaplan, 2002*).

Cardiac development and function strongly rely on myocardial wall integrity, which itself is modulated by cardiac tissue interactions as well as connections between cardiomyocytes (*Wu et al., 2009*; *Tian and Morrisey, 2012*; *Drechsler et al., 2013*; *Sojka et al., 2014*). Several studies have investigated how cardiac tissues including the endocardium, myocardium and epicardium communicate with each other to generate a correctly patterned and functioning organ (*Lee et al., 1995*; *Stainier et al., 1996*; *de la Pompa and Epstein, 2012*; *Ruiz-Villalba and Pérez-Pomares, 2012*; *Bressan et al., 2014*; *Wang et al., 2015*; *Rasouli and Stainier, 2017*; *Del Monte-Nieto et al., 2018*). While many signaling pathways including Notch (*Rones et al., 2000*; *High et al., 2009*; *D'Amato et al., 2016*), Wnt (*Marvin et al., 2001*; *Schneider and Mercola, 2001*), Retinoic acid

(*Stainier and Fishman, 1992*; *Keegan et al., 2005*), Hedgehog (*Washington Smoak et al., 2005*; *Wang et al., 2015*), Neuregulin/ErbB2 (*Lee et al., 1995*; *Meyer and Birchmeier, 1995*; *Crone et al., 2002*; *Lai et al., 2010*; *Rasouli and Stainier, 2017*), FGF (*Lavine et al., 2005*), and Angiopoietin-1/Tie2 (*Tachibana et al., 2005*; *Kim et al., 2018*) have been implicated in cardiac development, it is not clear how they interact to orchestrate this process, or how genes affected by cardiac function modulate cardiac formation.

Shear stress generated by blood flow can regulate hemodynamic responses that are vital for cardiovascular morphogenesis including blood vessel maturation (*Ehling et al., 2013*; *Nakajima and Mochizuki, 2017a*; *Nakajima et al., 2017b*), atrioventricular valve leaflet formation (*Slough et al., 2008*; *Vermot et al., 2009*; *Heckel et al., 2015*; *Steed et al., 2016a*; *Goddard et al., 2017*), and trabeculation (*Peshkovsky et al., 2011*; *Staudt et al., 2014*; *Samsa et al., 2015*; *Rasouli and Stainier, 2017*). However, the cellular and molecular mechanisms underlying this regulation are poorly understood.

Krüppel-like factors (KLFs) are transcriptional regulators that modulate cell growth and differentiation as well as tissue development. Some members of the KLF family, including KLF2, 4, 6, 10 and 15, have been implicated in cardiovascular development and function (*Lee et al., 2006*; *McConnell and Yang, 2010*; *Nayak et al., 2011*; *Dietrich et al., 2014*; *Renz et al., 2015*). *Klf2*, a flow responsive gene, is a well-known member of this family, and it is expressed in endothelial cells and lymphocytes (*Dekker, 2002*; *Cao et al., 2010*; *Wang et al., 2010*). $Klf2^{-/-}$ mice die between E12.5 and E14.5 due to hemorrhaging and cardiovascular failure (*Lee et al., 2006*; *Wu et al., 2008*). Since loss of *Klf2* function in mouse results in embryonic lethality, a detailed understanding of its role in cardiovascular development is still lacking. In order to complement studies in mouse and gain additional insights into the role of *klf2* in cardiovascular development, we used the zebrafish (*Danio rerio*), a model organism ideally suited for developmental genetic studies, especially those of the cardiovascular system (*Stainier and Fishman, 1994*; *Stainier et al., 1996*; *Beis et al., 2005*; *Chi et al., 2008*; *Staudt and Stainier, 2012*; *Collins and Stainier, 2016*).

Zebrafish *klf2a* and *klf2b* are known to be collectively orthologous to mammalian *Klf2* (*Oates et al., 2001*). Endothelial *klf2a* expression has been shown to be flow-dependent (*Parmar et al., 2006*), and its knockdown by antisense morpholinos was reported to cause pericardial edema, dysfunctional cardiac valves (*Vermot et al., 2009*) and defective haematopoietic stem cell development (*Wang et al., 2011*). However, unlike the aforementioned *klf2a* morpholino-induced cardiovascular defects, no obvious morphological phenotypes were observed in *klf2a*$^{sh317}$ mutant embryos (*Novodvorsky et al., 2015*), suggesting possible compensation (*Rossi et al., 2015*) by *klf2b* and/or other genes.

Taking a loss-of-function approach, we find that Klf2 is essential for myocardial wall integrity as loss of both *klf2a* and *klf2b* leads to a novel phenotype of cardiomyocyte extrusion to the abluminal side. Using high-resolution microscopy, we show that this extrusion is correlated with the mislocalization of N-Cadherin in *klf2* mutant cardiomyocytes. Taking a tissue-specific rescue approach, we find that endothelial Klf2 can maintain myocardial wall integrity and rescue *klf2* mutants to adulthood. Transcriptome analysis of *klf2* WT and mutant hearts led us to analyze Fgf signaling and we found that inhibition of Fgf signaling in wild-type (WT) animals could recapitulate the cardiomyocyte extrusion phenotype observed in *klf2* mutants. Overall, these data provide new insights into how an endocardially expressed gene can modulate myocardial wall integrity, and therefore further our knowledge about the complex endocardial-myocardial interactions underlying myocardial wall maturation.

## Results

### *klf2* mutants exhibit a cardiomyocyte extrusion phenotype

Given the role of *Klf2* in cardiovascular development in mouse (*Lee et al., 2006*), we used the Transcription Activator-Like Effector Nuclease (TALEN) technology (*Cermak et al., 2011*) to mutate both *klf2* paralogues in zebrafish (*Figure 1A–B* and *Figure 1—figure supplement 1A–B*), and identified a *klf2a* Δ10 allele (*bns11*) and a *klf2b* Δ8 allele (*bns12*) (*Kwon et al., 2016*). Notably, the predicted truncated Klf2a bns11 protein lacks the transrepression and zinc finger domains (*Figure 1A*), and the predicted Klf2b bns12 protein contains only part of the transactivation domain (*Figure 1B*).

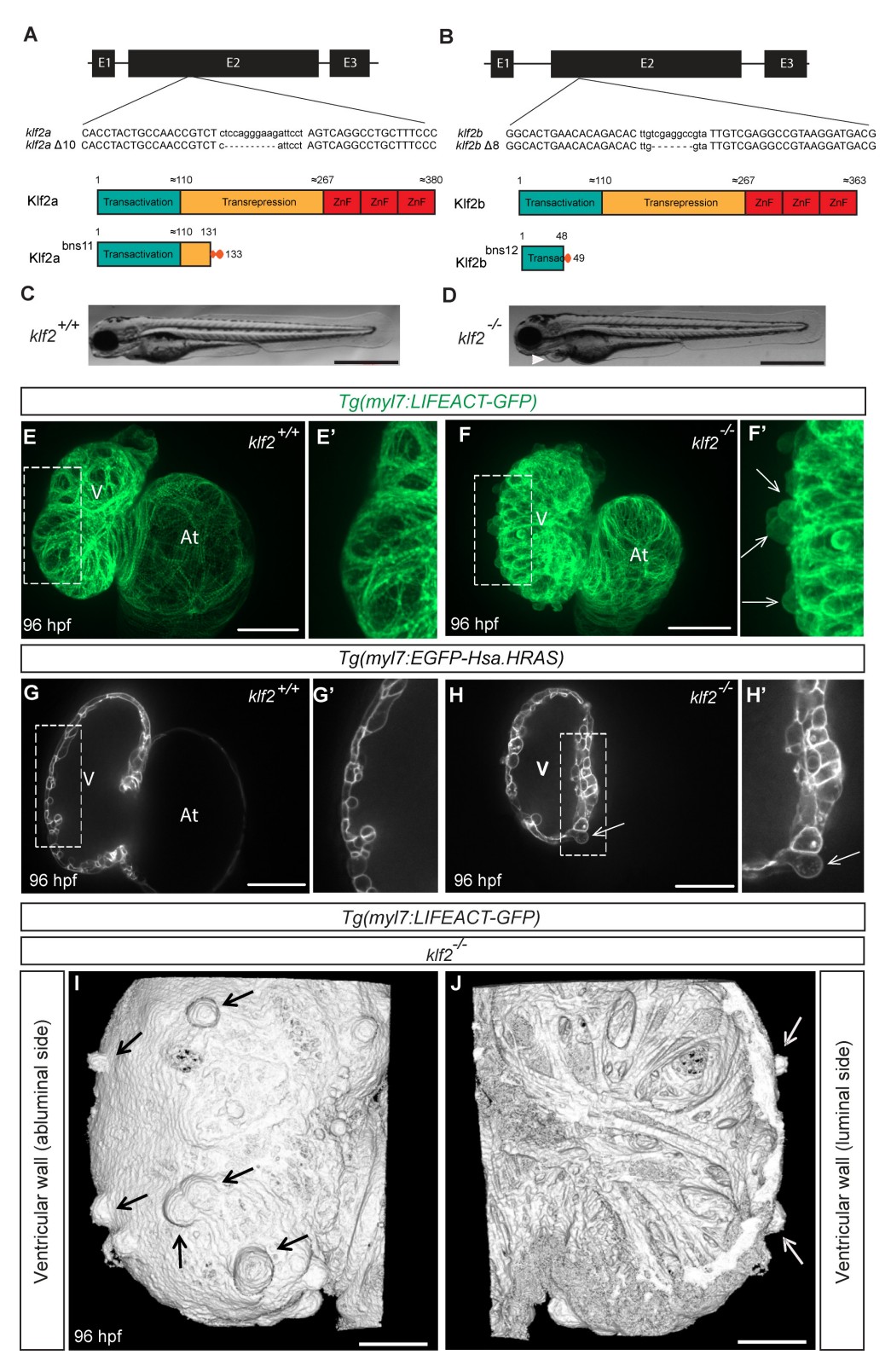

**Figure 1.** *klf2* mutants exhibit a cardiomyocyte extrusion phenotype. (A–B) Partial sequence alignment of *klf2a^bns11^* (A) and *klf2b^bns12^* (B) alleles with WT and schematics of their predicted protein products. (C–D) Representative brightfield images of a WT (C) and a *klf2a^bns11/bns11^*; *klf2b^bns12/bns12^* double mutant (hereafter referred to as *klf2* mutant) at 96 hpf (D); lateral views, anterior to the left; arrowhead points to pericardial edema. (E–F') Maximum intensity projections of 96 hpf *klf2* WT (E–E') and mutant (F–F') hearts; (G–H') Two-dimensional (2D) confocal images of 96 hpf *klf2* WT (G–G') and

*Figure 1 continued*

mutant (**H–H′**) hearts; ventricular outer curvature (dashed boxes) in (**E, F, G** and **H**) magnified in (**E′, F′, G′** and **H′**), respectively. (**I–J**) Three-dimensional reconstructions from confocal images of 96 hpf *klf2* mutant ventricular wall. Arrows point to extruding cardiomyocytes; V: ventricle; At: atrium; scale bars: 0.5 mm (**C–D**), 50 μm (**E–H**), 20 μm (**I–J**).

DOI: https://doi.org/10.7554/eLife.38889.002

The following figure supplements are available for figure 1:

**Figure supplement 1.** Identifying the *klf2a^bns11^* and *klf2b^bns12^* mutant alleles in the absence or presence of the *klf2-p2A-tdTomato* transgene.

DOI: https://doi.org/10.7554/eLife.38889.003

**Figure supplement 2.** Single *klf2a* or *klf2b* mutants do not exhibit any obvious phenotypes.

DOI: https://doi.org/10.7554/eLife.38889.004

**Figure supplement 3.** Characterizing additional cardiac phenotypes in *klf2* mutants.

DOI: https://doi.org/10.7554/eLife.38889.005

**Figure supplement 4.** Additional quantification of the cardiomyocyte extrusion phenotype in *klf2* mutants.

DOI: https://doi.org/10.7554/eLife.38889.006

Surprisingly, unlike the *klf2a* morphants which have been reported to exhibit pericardial edema and cardiac defects (*Vermot et al., 2009*), neither *klf2a^bns11^* nor *klf2b^bns12^* mutants exhibit gross morphological defects (*Figure 1—figure supplement 2A–C*). Close examination revealed no obvious cardiovascular defects in *klf2a* or *klf2b* single mutants (*Figure 1—figure supplement 2D–P*) which can survive to become fertile adults. Quantitative PCR (qPCR) analysis of *klf2a*, *klf2b* and *klf4a* mRNA levels shows that *klf2a* transcript levels are significantly reduced in *klf2a^bns11^* mutants compared to their WT siblings, suggesting nonsense-mediated decay. In addition, the levels of *klf2b* and *klf4a* mRNA are increased in *klf2a^bns11^* mutants compared to their WT siblings (*Figure 1—figure supplement 2Q*), indicating that they could be compensating for the loss of Klf2a function. Similarly, *klf2a* and *klf4a* mRNA levels are increased in *klf2b^bns12^* mutants compared to their WT siblings (*Figure 1—figure supplement 2R*). We thus decided to generate *klf2a^bns11^*; *klf2b^bns12^* double mutants (hereafter referred to as *klf2* mutants). Unlike the single mutants, most double mutants exhibit pericardial edema by 96 hours post fertilization (hpf) (*Figure 1C–D* and *Figure 1—figure supplement 3A*). Abnormal atrioventricular (AV) canal elongation was observed at 48 hpf in approximately 10% of *klf2* mutants (3/26). By 80 hpf, AV valve formation was clearly impaired in approximately 50% of *klf2* mutants (17/36) as the canal appeared more elongated than in WT, and the multi-layering process leading to leaflet formation was also compromised in these animals (*Figure 1—figure supplement 3B–E*), consistent with previous observations in *klf2a* morphants (*Vermot et al., 2009*). Although the vascular network appeared unaffected in *klf2* mutants, most of them died by 2 weeks post-fertilization and none survived to adulthood (*Figure 1—figure supplement 3F–H*). Upon closer examination of their hearts at 96 hpf, we found that approximately 90% of *klf2* mutants exhibit a previously unreported phenotype of cardiomyocyte extrusion toward the abluminal side (69/75 mutant hearts) (*Figure 1E–J*). This phenotype first appears around 82 hpf in approximately 15% of *klf2* mutants (11/67 mutant hearts), and the number of extruding cardiomyocytes, which are mostly located in the outer curvature of the ventricle, is variable and increases over time (*Figure 1—figure supplement 4A–B*). We also found that the size and circularity of *klf2* mutant cardiomyocytes was significantly affected starting at 96 hpf (*Figure 1—figure supplement 4C–D*), possibly suggesting that the abnormally shaped cardiomyocytes represent an intermediate population on their way to extrusion.

## Cardiac contractility is required for cardiomyocyte extrusion

Cardiac contractility/blood flow is vital for cardiac morphogenesis and function (*Sehnert et al., 2002*; *Berdougo et al., 2003*; *Peshkovsky et al., 2011*; *Samsa et al., 2015*; *Collins and Stainier, 2016*; *Rasouli and Stainier, 2017*). We found that around 60 percent of *klf2* mutants exhibit impaired cardiac contractility at 96 hpf. In order to test whether the cardiomyocyte extrusion phenotype was secondary to impaired contractility, we first checked whether it could be induced in WT animals by modulating contractility with *amhc* morpholinos (MOs) which completely inhibit atrial contractility and cause secondary effects on ventricular contractility (*Berdougo et al., 2003*), as well as *tnnt2a* MOs which block both atrial and ventricular contractility (*Sehnert et al., 2002*). We observed that impaired contractility could not induce cardiomyocyte extrusion in WT animals (*Figure 2—*

figure supplement 1), indicating that this phenotype is not secondary to impaired contractility. However, injecting *tnnt2a* MOs into *klf2* mutants at the one-cell stage blocked cardiomyocyte extrusion (**Figure 2A**). In addition, short-term inhibition of cardiac contractility by BDM treatment from 100 to 102 hpf could also block cardiomyocyte extrusion (4/5 hearts) (**Figure 2B**). Overall, these data indicate that cardiac contractility is required for cardiomyocyte extrusion in *klf2* mutants, and that this extrusion is reversible.

## Cardiomyocyte extrusion correlates with N-cadherin mislocalization but not with cardiomyocyte death or proliferation

Cell death, crowding and loss of cell adhesion molecules have been implicated in epithelial cell extrusion (*Rosenblatt et al., 2001*; *Semenza, 2008*; *Chien et al., 2008*; *Eisenhoffer et al., 2012*; *Marinari et al., 2012*; *Kuipers et al., 2014*). In order to investigate the cause of the cardiomyocyte extrusion phenotype in *klf2* mutants, we first performed Acridine Orange staining and found that the extruding cardiomyocytes in *klf2* mutants did not appear to be dying (**Figure 3A–D**). To investigate cardiomyocyte proliferation, we used the *Tg(myl7:mVenus-gmnn)* line (*Jiménez-Amilburu et al., 2016*; *Uribe et al., 2018*) which allows one to visualize actively proliferating cardiomyocytes in green, and found that while the rate of atrial cardiomyocyte proliferation appeared unchanged in *klf2* mutants compared to their WT siblings, the levels of ventricular cardiomyocyte

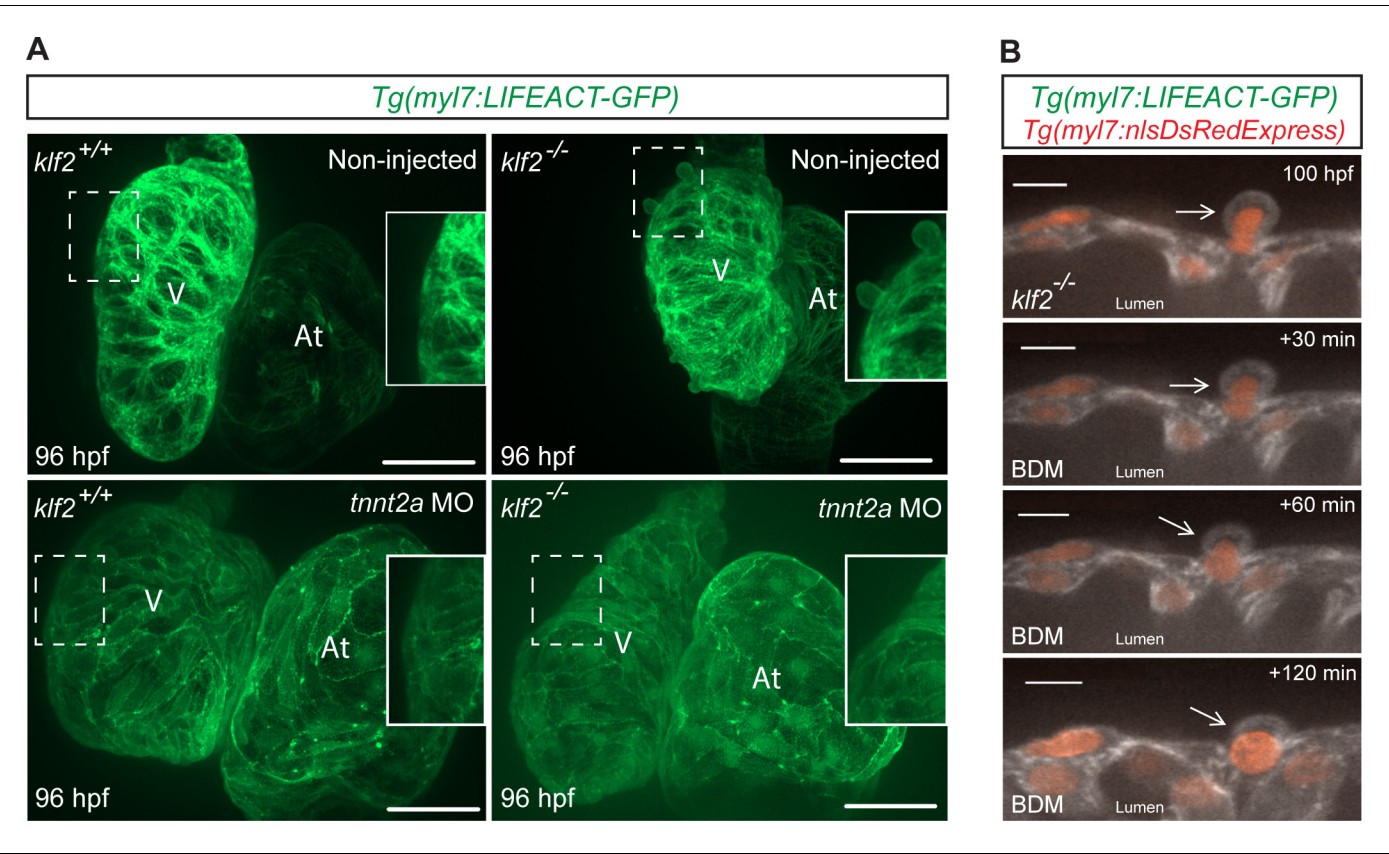

**Figure 2.** Cardiac contractility is required for cardiomyocyte extrusion. (**A**) Maximum intensity projections of confocal z-stacks of 96 hpf *klf2* WT and mutant hearts, non-injected or injected with *tnnt2a* MO at the one-cell stage; part of ventricular outer curvature (dashed boxes) magnified on the right side of each panel. (**B**) Time-lapse 2D confocal images of a *klf2* mutant heart during BDM treatment. Arrows point to an extruding cardiomyocyte returning to the compact layer upon inhibiting contraction; V: ventricle; At: atrium; scale bars: 50 μm (**A**), 10 μm (**B**).
DOI: https://doi.org/10.7554/eLife.38889.007
The following figure supplement is available for figure 2:

**Figure supplement 1.** Cardiomyocyte extrusion from the compact layer is not caused by impaired contractility.
DOI: https://doi.org/10.7554/eLife.38889.008

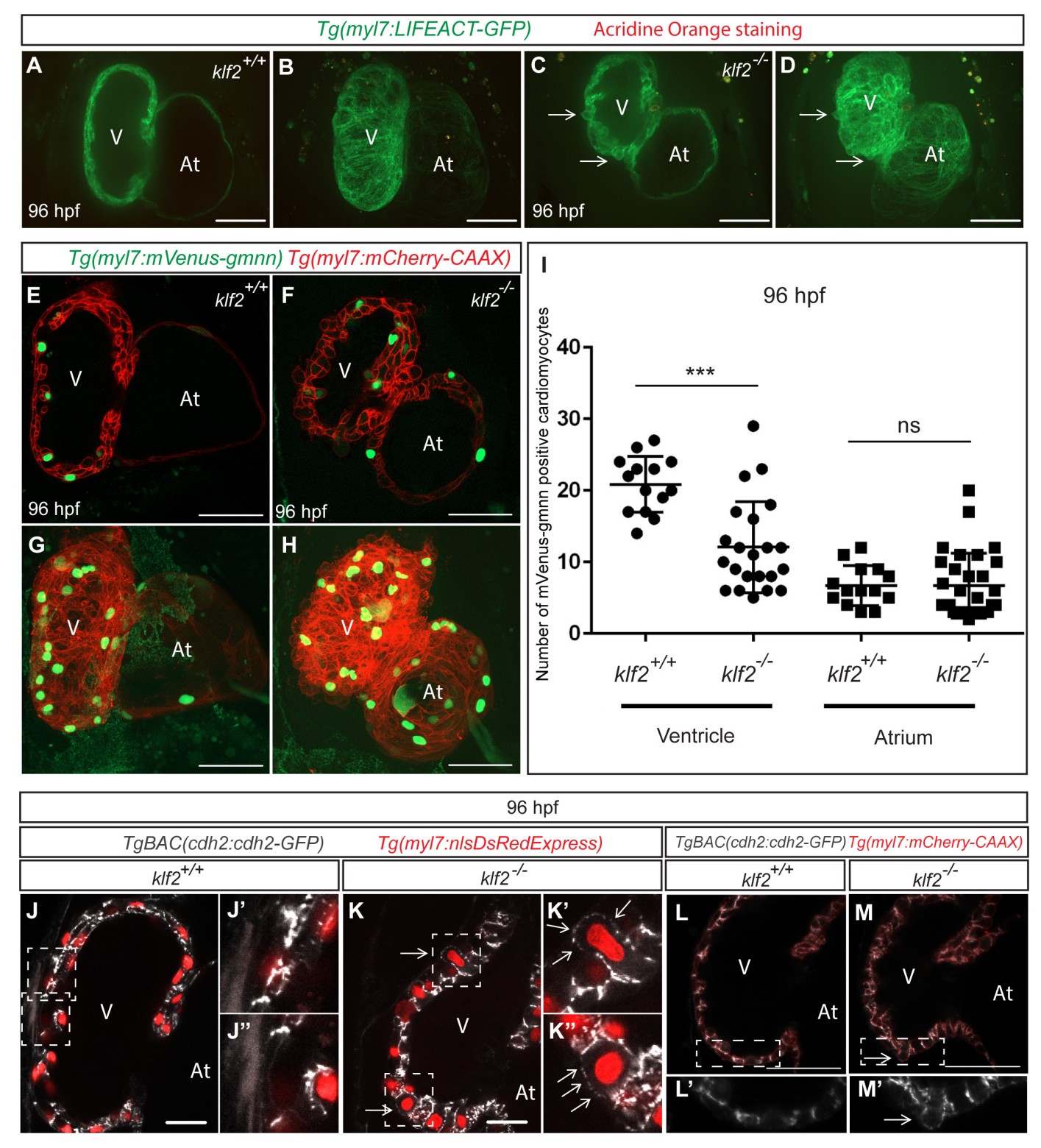

**Figure 3.** Cardiomyocyte extrusion correlates with N-cadherin mislocalization but not with cardiomyocyte death or proliferation. (**A–D**) 2D (mid-sagittal sections) (**A and C**) and maximum intensity projections of confocal z-stacks (**B and D**) of 82 hpf *klf2* WT (**A–B**) and mutant (**C–D**) hearts stained with Acridine Orange to visualize cell death; arrows point to extruding cardiomyocytes. (**E–H**) Confocal images of 96 hpf *klf2* WT (**E and G**) and mutant (**F and H**) hearts to visualize cardiomyocyte proliferation. (**I**) Number of mVenus-gmnn positive ventricular and atrial cardiomyocytes in 96 hpf *klf2* WT and mutant hearts; dots represent individual hearts; values represent means ±SEM; \***p≤0.001, ns (not significant), by Student's *t*-test. (**J–K''**) Mid-sagittal

*Figure 3 continued on next page*

*Figure 3 continued*
confocal sections of 96 hpf *klf2* WT (J) and mutant (K) hearts. Higher magnification images of the outer curvature of the ventricular wall (white dashed boxes) in (J) and (K) are shown in (J'), (J''), (K') and (K''); arrows point to ectopic accumulation of Cdh2-EGFP proteins on the apical side of cardiomyocytes. (L–M') 2D confocal views of 96 hpf *klf2* WT (L) and mutant (M) hearts. Magnified images of dashed boxes in (L) and (M) are shown in (L') and (M'), respectively. Arrows point to mislocalized Cdh2-GFP on the apical side of cardiomyocytes; V: ventricle, At: atrium; scale bars, 50 μm.
DOI: https://doi.org/10.7554/eLife.38889.009
The following figure supplements are available for figure 3:

**Figure supplement 1.** Downregulation of *cdh2* leads to cardiomyocyte extrusion.
DOI: https://doi.org/10.7554/eLife.38889.010
**Figure supplement 2.** The extruding cardiomyocytes appear polarized in the apicobasal axis.
DOI: https://doi.org/10.7554/eLife.38889.011
**Figure supplement 3.** Reduction of the cardiac jelly is not obviously affected in *klf2* mutants.
DOI: https://doi.org/10.7554/eLife.38889.012
**Figure supplement 4.** The epicardium is affected in *klf2* mutants.
DOI: https://doi.org/10.7554/eLife.38889.013

proliferation were considerably reduced (*Figure 3E–I*), suggesting that crowding is not the cause of the cardiomyocyte extrusion phenotype in *klf2* mutants. These observations are consistent with recent reports that inhibition of cardiomyocyte proliferation in zebrafish can reduce the total number of cardiomyocytes while not leading to their extrusion (*Liu et al., 2010*; *Uribe et al., 2018*).

It has been reported that N-cadherin (Cdh2), an adhesion molecule, is highly enriched in mouse and zebrafish cardiomyocytes (*Kostetskii et al., 2001*; *Bagatto et al., 2006*; *Cherian et al., 2016*). We thus examined the distribution of Cdh2-EGFP molecules using the *TgBAC(cdh2:cdh2-EGFP)* line (*Revenu et al., 2014*), and found that while they are highly enriched on the lateral sides of WT cardiomyocytes at 96 hpf, they have become mislocalized from the lateral to the apical side of cardiomyocytes in *klf2* mutants (*Figure 3J–M'*), suggesting that cardiomyocyte adhesion is affected. In addition, knockdown of N-cadherin by injecting *cdh2* MO at the one-cell stage can also cause a cardiomyocyte extrusion phenotype in WT hearts (*Figure 3—figure supplement 1*). Overall, these data suggest that cardiomyocyte extrusion correlates with the mislocalization of N-cadherin on their membranes but not with cardiomyocyte cell death or proliferation.

## The extruding cardiomyocytes appear polarized in the apicobasal axis

It has recently been shown that cardiomyocytes in the compact layer are polarized in their apicobasal axis whereas during the trabeculation process, they lose this polarity as they delaminate (*Jiménez-Amilburu et al., 2016*). In order to further investigate the extruding cardiomyocytes, we checked their apicobasal polarity by generating a myocardial-specific apical transgenic line, *Tg(−0.2myl7:tdtomato-podxl)*, in which the apical protein podocalyxin is fused with tdTomato, and found that at 96 hpf, extruding cardiomyocytes do not lose their apical polarity (*Figure 3—figure supplement 2*).

## Reduction of the cardiac jelly is not obviously affected in *klf2* mutants

Reduction of the cardiac jelly, a gelatinous matrix between the endocardial and myocardial layers, is an important process during cardiac morphogenesis (*Brutsaert et al., 1996*; *Stankunas et al., 2008*; *Bowers and Baudino, 2010*; *Tian and Morrisey, 2012*; *Rasouli and Stainier, 2017*; *Grassini et al., 2018*). In order to investigate whether reduction of the cardiac jelly was affected in *klf2* mutants, we used the *TgBAC(etv2:EGFP)* (*Proulx et al., 2010*) and *Tg(myl7:mCherry-CAAX)* (*Uribe et al., 2018*) lines to mark the endocardium and myocardium in green and red, respectively. Utilizing high-resolution confocal microscopy at 96 hpf, we observed that there was no significant difference in terms of cardiac jelly thickness between *klf2* WT and mutant animals (*Figure 3—figure supplement 3*).

## The epicardium is affected in *klf2* mutants

The epicardium, the outer layer of vertebrate hearts, is known to be involved in the nourishment of the underlying myocardium (*Peralta et al., 2014*), the formation of coronaries (*Ruiz-Villalba and Pérez-Pomares, 2012*; *González-Rosa et al., 2012*), and tissue regeneration after cardiac injury (*Schlueter and Brand, 2012*; *González-Rosa et al., 2012*; *Wang et al., 2015*). In order to

investigate the behavior of epicardial cells during early zebrafish heart development, we used high-resolution microscopy to image hearts in *Tg(myl7:mCherry-CAAX);TgBAC(tcf21:NLS-EGFP)* (*Wang et al., 2015*) animals, in which myocardial and epicardial cells are labeled in red and green, respectively. At 55 hpf, we observed a few *tcf21*-positive cells attaching to the abluminal side of the ventricle, which was fully covered by 96 hpf (*Figure 3—figure supplement 4A–H*), in agreement with previous studies (*Virágh and Challice, 1981*; *Peralta et al., 2013*; *Peralta et al., 2014*; *Wang et al., 2015*). At 96 hpf, we observed fewer *tcf21*-positive cells covering the *klf2* mutant hearts compared to their WT siblings, and also a sparser distribution (*Figure 3—figure supplement 4I–M*), suggesting defective epicardial-myocardial interactions in the absence of Klf2 function. In order to investigate whether incomplete epicardial coverage could contribute to the cardiomyocyte extrusion phenotype observed in *klf2* mutants, we used a genetic ablation approach to remove epicardial cells in *Tg(myl7:EGFP-Has.HRAS)* (*D'Amico et al., 2007*); *TgBAC(tcf21:mCherry-NTR)* (*Wang et al., 2015*) animals using metronidazole (MTZ) treatment from 48 to 96 hpf. We found that this epicardial ablation approach could also induce cardiomyocyte extrusion; however, the number of extruding cardiomyocytes after epicardial ablation was far lower than that observed in *klf2* mutants (*Figure 3—figure supplement 4N–R*).

## Klf2 functions cell non-autonomously to maintain the integrity of the myocardial wall

To investigate where Klf2 function is required to maintain the integrity of the myocardial wall, we generated chimeric hearts by cell transplantation and examined the behavior of WT cardiomyocytes in *klf2* mutant hearts, and vice versa (*Figure 4*). We first transplanted *Tg(myl7:MKATE-CAAX)* WT cells into *Tg(myl7:LIFEACT-GFP); klf2*<sup>+/?</sup> or *klf2*<sup>-/-</sup> hosts. Using confocal microscopy, we found that at 96 hpf some WT donor-derived cardiomyocytes were extruding from *klf2*<sup>-/-</sup> but not *klf2*<sup>+/?</sup> hearts (*Figure 4A–E*; n = 7 hearts). We also performed the converse experiment by transplanting *Tg(myl7: LIFEACT-GFP); klf2*<sup>+/?</sup> or *klf2*<sup>-/-</sup> cells into WT hosts and observed no extruding *klf2*<sup>-/-</sup> cardiomyocytes (*Figure 4F–J*; n = 9 hearts). These data indicate that Klf2 function is not required in cardiomyocytes to maintain the integrity of the myocardial wall. *klf2b* overexpression in endothelial cells can rescue the *klf2* mutant cardiomyocyte extrusion phenotype

In zebrafish, as in mouse, *klf2* is expressed in the endocardium (*Lee et al., 2006*; *Vermot et al., 2009*). Similar to *klf2a*, *klf2b* also becomes expressed in the zebrafish heart by 36 hpf (*Figure 5—figure supplement 1*). To investigate where Klf2 function is required to prevent the cardiomyocyte extrusion phenotype observed in *klf2* mutants (*Figure 5A–B and and E–F*), we generated a *Tg(fli1a: klf2b-p2A-tdTomato)* line to specifically overexpress *klf2b* in all endothelial cells including the endocardium (*Figure 5C–D and and G–H*). We also generated *Tg(myl7:klf2b-p2A-tdTomato)* and *Tg (myl7:klf2a-p2A-tdTomato)* lines to overexpress *klf2a* and *klf2b* in cardiomyocytes (*Figure 5I–P* and *Figure 1—figure supplement 1C–D*). Unlike endothelial *klf2b* overexpression, myocardial *klf2a* and *klf2b* overexpression caused pericardial edema and impaired trabeculation in about forty percent of the F1 and F2 animals when observed at 96 hpf. These cardiac defects became milder beyond the F2 generation. Notably, we observed that endothelial, but not myocardial, *klf2b* overexpression rescued the pericardial edema as well as the cardiomyocyte extrusion phenotype observed in *klf2* mutant hearts (*Figure 5A–P*), indicating the importance of endothelial/endocardial *klf2* expression in maintaining myocardial wall integrity. We also quantified the number of proliferative cardiomyocytes in *klf2* WT and mutant animals overexpressing *klf2b* in their endothelium and found that the reduction of ventricular cardiomyocyte proliferation observed in *klf2* mutants was rescued by endothelial *klf2b* overexpression (*Figure 5—figure supplement 2*). Interestingly, endothelial *klf2b* overexpression also rescued the epicardial defects observed in *klf2* mutants (*Figure 5—figure supplement 3*). Overall, these data indicate that endothelial/endocardial *klf2* is required for myocardial wall integrity by regulating endocardial-myocardial-epicardial interactions. Although all *klf2* mutant larvae die, we were able to rescue almost sixty percent of them to become adults using this endothelial rescue approach (*Figure 5—figure supplement 4A–B*). Dissecting the adult hearts, we did not observe any significant differences between *klf2* WT and rescued mutant hearts, further suggesting the importance of endothelial *klf2* expression in maintaining myocardial wall integrity (*Figure 5Q–R'* and *Figure 5—figure supplement 4C–F*).

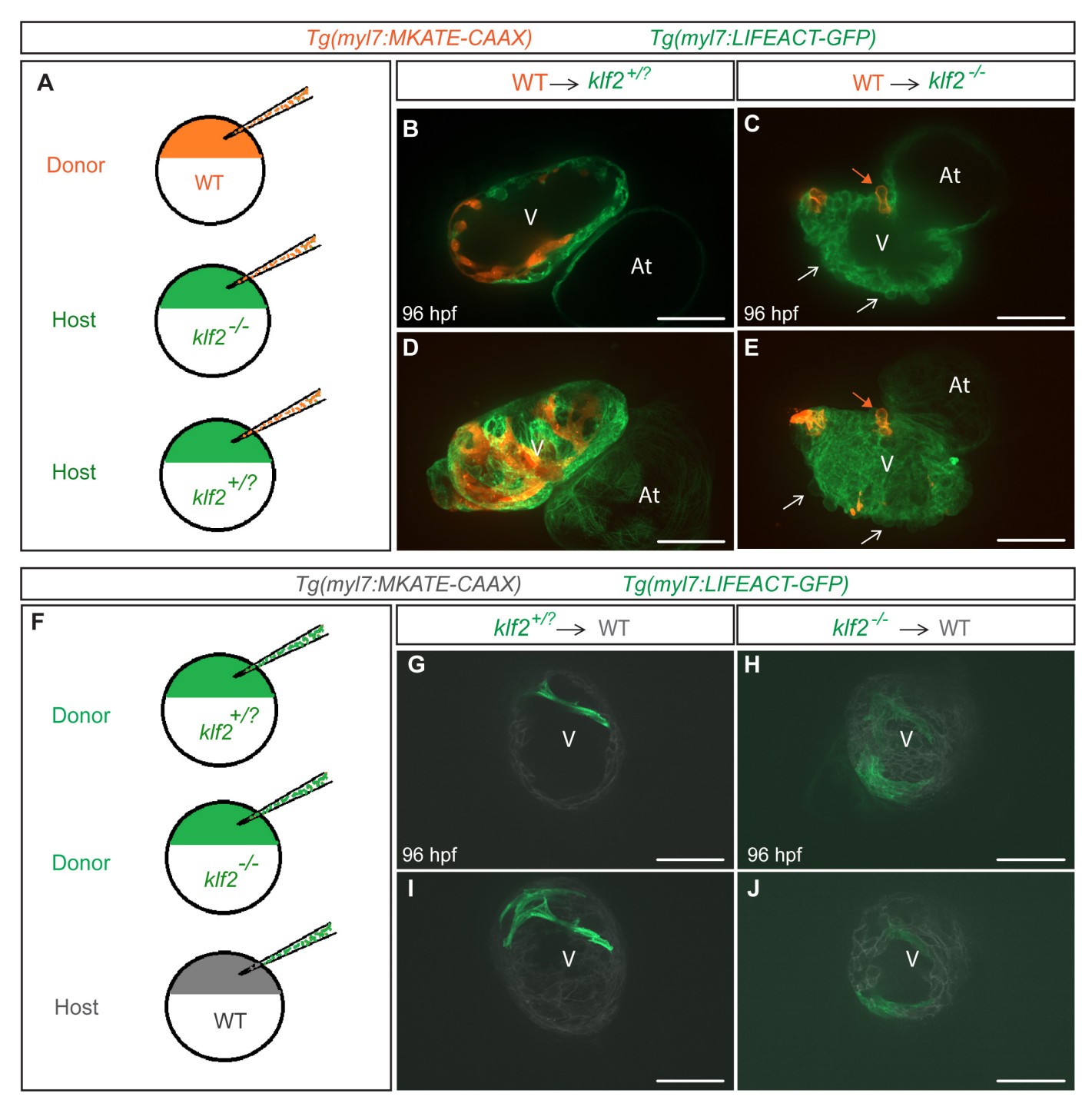

**Figure 4.** Klf2 functions cell non-autonomously to maintain the integrity of the myocardial wall. (A) Schematic representation of the experiment shown in (B–E). (B–E) Transplantation of *Tg(myl7:MKATE-CAAX); klf2*$^{+/+}$ donor cells into *Tg(myl7:LIFEACT-GFP); klf2*$^{+/?}$ (B and D) or *klf2*$^{-/-}$ (C and E) hosts shown at 96 hpf; white arrows point to *klf2*$^{-/-}$ extruding cardiomyocytes in *klf2*$^{-/-}$ heart, orange arrows point to *klf2*$^{+/+}$ extruding cardiomyocytes in *klf2*$^{-/-}$ hearts; maximum intensity projections of confocal z-stacks of hearts in (B) and (C) are shown in (D) and (E), respectively. (F) Schematic representation of the experiment shown in (G-J). (G–J) Transplantation of *Tg(myl7:LIFEACT-GFP); klf2*$^{+/?}$ (G and I) or *klf2*$^{-/-}$ (H and J) donor cells into *Tg(myl7:MKATE-CAAX); klf2*$^{+/+}$ hosts shown at 96 hpf; maximum intensity projections of confocal z-stacks of hearts in (G) and (H) are shown in (I) and (J), respectively. V: ventricle, At: atrium; scale bars, 50 μm.

DOI: https://doi.org/10.7554/eLife.38889.014

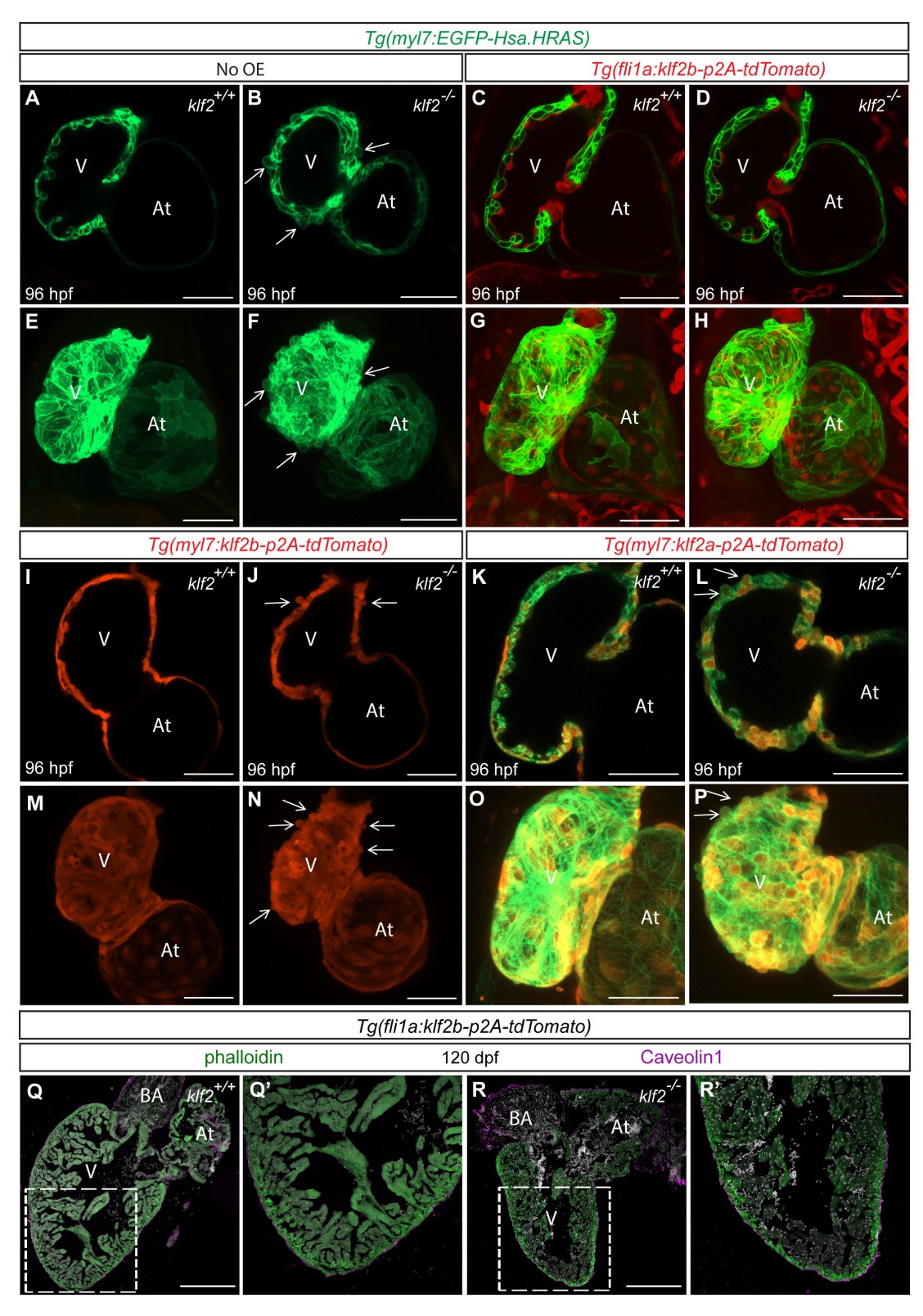

**Figure 5.** *klf2b* overexpression in endothelial cells can rescue the *klf2* mutant cardiomyocyte extrusion phenotype. (A–P) Endothelial- and myocardial-specific overexpression of *klf2a* or *klf2b* in *klf2* WT and mutant hearts. Endothelial overexpression of *klf2b* (C–D, G–H). Myocardial overexpression of *klf2a* (K–L, O–P) or *klf2b* (I–J, M–N); maximum intensity projections of hearts in (A–D) and (I–L) are shown in (E–H) and (M–P), respectively. (Q–R') Immunostaining of adult *klf2* WT (Q–Q') and rescued mutant (R–R') heart sections for Caveolin1 to label epicardial cells and phalloidin for overall

*Figure 5 continued on next page*

*Figure 5 continued*

myocardial structure; magnified images of dashed boxes in (**Q**) and (**R**) are shown in (**Q'**) and (**R'**), respectively; arrows point to extruding cardiomyocytes; V: ventricle, At: atrium; scale bars: 50 μm (**A–P**), 300 μm (**Q–R'**).

DOI: https://doi.org/10.7554/eLife.38889.015

The following figure supplements are available for figure 5:

**Figure supplement 1.** *klf2b* expression at early developmental stages .

DOI: https://doi.org/10.7554/eLife.38889.016

**Figure supplement 2.** The cardiomyocyte proliferation defect in *klf2* mutants is rescued by endothelial *klf2b* overexpression.

DOI: https://doi.org/10.7554/eLife.38889.017

**Figure supplement 3.** The epicardial defect in *klf2* mutants is rescued by endothelial *klf2b* overexpression.

DOI: https://doi.org/10.7554/eLife.38889.018

**Figure supplement 4.** *klf2* mutants rescued by endothelial *klf2b* overexpression can survive to adulthood.

DOI: https://doi.org/10.7554/eLife.38889.019

## Increased Retinoic Acid signaling or inhibition of Hedgehog signaling does not cause a cardiomyocyte extrusion phenotype

To further understand how endothelial cells regulate myocardial wall integrity, we performed microarray analysis of 96 hpf *klf2* WT and mutant hearts. Our transcriptome data show that *aldh1a2* and *cyp26b1*, the highest expressing *cyp26* gene in 96 hpf hearts, are respectively up- and down-regulated in *klf2* mutant hearts (*Figure 6—source data 1*), suggesting an increase in retinoic acid (RA) levels. It has recently been reported that excess levels of RA in *cyp26a1* and *cyp26c1* morphants lead to the failure of second heart field progenitors to join the outflow tract as well as the extrusion of first heart field-derived cardiomyocytes from the ventricle at 48 hpf (*Rydeen and Waxman, 2016*). In order to test whether increased RA signaling causes cardiomyocyte extrusion at later stages, we treated WT zebrafish larvae with RA from 74 to 96 hpf but did not observe any extruding cardiomyocytes (*Figure 6—figure supplement 1*). The later onset of RA treatment used in our experiments likely explains the different outcome from that observed by *Rydeen and Waxman, 2016*. We also found that Hedgehog (Hh) signaling was downregulated in *klf2* mutant hearts (*Figure 6—figure supplement 2*, *Supplementary file 2* and *Figure 6—source data 2*); however, inhibition of this pathway in WT zebrafish from 75 to 96 hpf using Cyclopamine, a Hh signaling pathway inhibitor, did not cause cardiomyocyte extrusion (*Figure 6—figure supplement 3*). Together, these data indicate that by itself increased RA signaling or inhibition of Hh signaling does not cause the cardiomyocyte extrusion phenotype observed in *klf2* mutants.

## Inhibition of Fgfr signaling can lead to cardiomyocyte extrusion in WT animals

Fgf signaling has been implicated in the formation of the compact myocardial layer as well as in cardiomyocyte differentiation and proliferation (*Reifers et al., 2000*; *Lavine et al., 2005*; *Marques et al., 2008*; *Tirosh-Finkel et al., 2010*; *Pradhan et al., 2017*). Transcriptomic analysis of *klf2* WT and mutant hearts shows that Fgf signaling is affected in *klf2* mutants (*Figure 6—figure supplement 2*, *Figure 6—source data 3* and *Supplementary file 2*); follow up qPCR analysis shows that several *fgf* receptor and ligand genes are expressed at reduced levels in *klf2* mutant compared to WT hearts (*Figure 6—figure supplement 4*). To test the role of Fgf signaling in myocardial wall integrity, we treated *Tg(myl7:EGFP-Has.HRAS)* larvae from 75 to 96 hpf with SU5402, a FGF receptor inhibitor, and observed cardiomyocyte extrusion (7/9 hearts; *Figure 6A–D*), as well as mislocalization of Cdh2-EGFP from the lateral to the apical side of cardiomyocytes (10/11 hearts; *Figure 6—figure supplement 5*). In addition, global overexpression of dominant negative Fgfr1, using the *Tg(hsp70:dn-fgfr1-EGFP)* line (*Lee et al., 2005*), could also cause cardiomyocyte extrusion (9/13 hearts; *Figure 6E–H*); however, the number of extruding cardiomyocytes in these animals was less than that observed in *klf2* mutants or SU5402 treated animals (*Figure 6—figure supplement 6A*). While, this cardiomyocyte extrusion phenotype appeared in only a few SU5402 treated hearts by 82 hpf (7/36 hearts), the number of extruding cardiomyocytes increased over time, as in *klf2* mutants (*Figure 6—figure supplement 6B–D*). These data indicate that inhibition of Fgfr signaling can cause cardiomyocyte extrusion, as in *klf2* mutants. We also treated WT and *klf2a*^+/-^; *klf2b*^-/-^ larvae with DMSO or

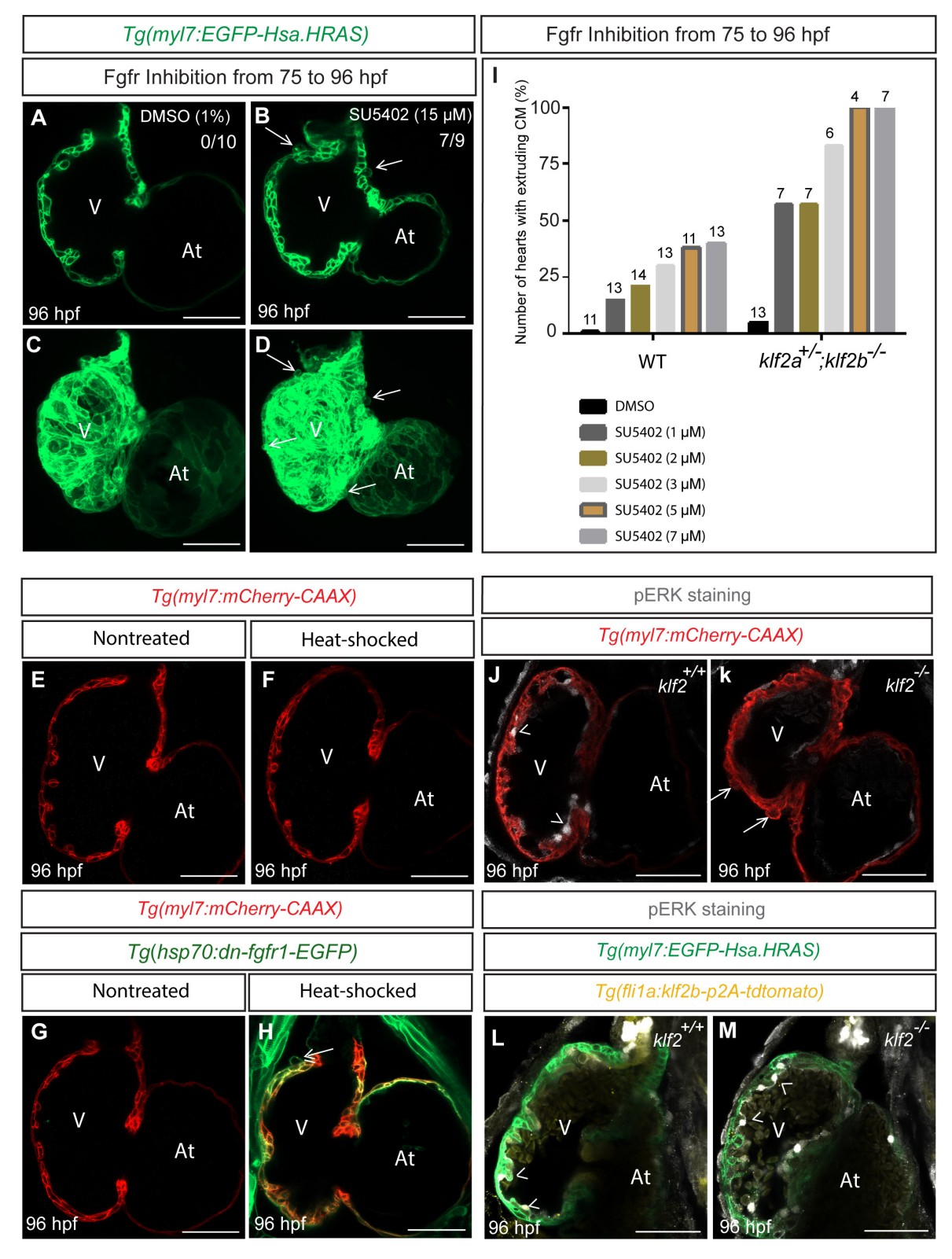

**Figure 6.** Inhibition of Fgfr signaling can lead to cardiomyocyte extrusion in WT animals. (**A–D**) Confocal images of 96 hpf hearts; WT animals treated with DMSO as a control or FGFR inhibitor (SU5402) from 75 to 96 hpf; maximum intensity projections of hearts in (**A**) and (**D**) are shown in (**C**) and (**D**) respectively; arrows point to extruding cardiomyocytes. (**E–H**) 75 hpf Tg(*myl7: mCherry-CAAX*) (**E–F**) or Tg(*hsp70:dn-fgfr1-EGFP*);Tg(*myl7: mCherry-CAAX*) (**G–H**) animals were heat-stressed at 39°C for 1 hr (**F and H**) and their hearts imaged at 96 hpf; arrow in (**H**) points to an extruding cardiomyocyte

*Figure 6 continued on next page*

*Figure 6 continued*

(n = 9/13 hearts). (I) *klf2a*$^{+/-}$; *klf2b*$^{-/-}$ animals are more likely than WT siblings to exhibit cardiomyocyte extrusion upon Fgfr inhibition; number of treated larvae for each condition is shown above the individual columns. (J–K) Hearts of 96 hpf *Tg(myl7: mCherry-CAAX); klf2* $^{+/+}$ *or klf2* $^{-/-}$ animals immunostained for pERK. (L–M) Hearts of 96 hpf *Tg(fli1a:klf2b-p2A-tdTomato);Tg(myl7:EGFP-Hsa.HRAS); klf2* $^{+/+}$ *or klf2* $^{-/-}$ animals immunostained for pERK. Arrows and arrowheads point to extruding cardiomyocytes and pERK positive endocardial cells, respectively; V: ventricle, At: atrium; scale bars, 50 μm.

DOI: https://doi.org/10.7554/eLife.38889.020

The following source data and figure supplements are available for figure 6:

**Source data 1.** Upregulation of *aldh1a2* and downregulation of *cyp26b1* in *klf2* mutant hearts compared to wild-type.
DOI: https://doi.org/10.7554/eLife.38889.030

**Source data 2.** Hedgehog signaling is affected in *klf2* mutant hearts.
DOI: https://doi.org/10.7554/eLife.38889.031

**Source data 3.** *fgf* ligand and receptor genes are downregulated in *klf2* mutant hearts.
DOI: https://doi.org/10.7554/eLife.38889.032

**Figure supplement 1.** Increased Retinoic Acid signaling does not cause cardiomyocyte extrusion.
DOI: https://doi.org/10.7554/eLife.38889.021

**Figure supplement 2.** Broad GSEA enrichment plots of selected down-regulated gene sets.
DOI: https://doi.org/10.7554/eLife.38889.022

**Figure supplement 3.** Inhibition of Hedgehog signaling does not cause cardiomyocyte extrusion.
DOI: https://doi.org/10.7554/eLife.38889.023

**Figure supplement 4.** mRNA levels of *fgf* ligand and receptor genes in WT and *klf2* mutant hearts.
DOI: https://doi.org/10.7554/eLife.38889.024

**Figure supplement 5.** Inhibition of Fgfr signaling can lead to Cdh2-GFP mislocalization in cardiomyocytes.
DOI: https://doi.org/10.7554/eLife.38889.025

**Figure supplement 6.** Additional quantification of the cardiomyocyte extrusion phenotype upon Fgfr inhibition.
DOI: https://doi.org/10.7554/eLife.38889.026

**Figure supplement 7.** Fgf signaling is required for ERK phosphorylation in endocardial cells.
DOI: https://doi.org/10.7554/eLife.38889.027

**Figure supplement 8.** Single cell graphs of *fgf* receptor and ligand genes expressed in zebrafish embryonic endothelium and heart.
DOI: https://doi.org/10.7554/eLife.38889.028

**Figure supplement 9.** mRNA levels of *fgf* ligand and receptor genes in WT and *npas4l* mutant hearts.
DOI: https://doi.org/10.7554/eLife.38889.029

---

increasing concentrations of SU5402 and found that *klf2a*$^{+/-}$; *klf2b*$^{-/-}$ animals were more sensitive to Fgfr signaling inhibition, that is more prone to exhibit cardiomyocyte extrusion compared to WT (*Figure 6I*). We further treated WT larvae with SU5402 from 75 to 96 hpf and then stained their hearts with a pERK antibody. Our data show that Fgfr inhibition reduced pERK immunostaining (*Figure 6—figure supplement 7*), in agreement with previous reports (*Shinya et al., 2001*; *Calmont et al., 2006*). We also checked pERK immunostaining after *dnfgfr1* overexpression and observed a reduction, although milder than that observed in SU5402-treated hearts (*Figure 6—figure supplement 7*). We further found that pERK immunostaining was significantly reduced in endocardial cells of *klf2* mutants compared to WT (*Figure 6J and K*), and that it was rescued by endothelial *klf2b* overexpression (*Figure 6L and M*). Overall, these data suggest that Fgf signaling is required downstream of Klf2 to maintain myocardial wall integrity. By examining single-cell RNA sequencing data of early zebrafish embryos (https://kleintools.hms.harvard.edu/paper_websites/wagner_zebrafish_timecourse2018/mainpage.html; *Wagner et al., 2018*), we found that most *fgf* receptor genes and many *fgf* ligand genes are expressed in the zebrafish heart (*Figure 6—figure supplement 8*). Because the level of expression of most of these genes is rather low, and possibly challenging to detect by in situ hybridization, we performed qPCR analysis in *cloche/npas4l* mutant hearts which lack all endocardial cells (*Stainier et al., 1995*; *Reischauer et al., 2016*). As expected, *myl7* expression levels were unaffected in *npas4l* mutant hearts compared to WT while *fli1a* levels were significantly reduced (*Figure 6—figure supplement 9A*). Expression levels of all *fgf* receptor genes were slightly increased in *npas4l* mutants (*Figure 6—figure supplement 9B*), suggesting that they are not expressed in the endocardium. Analyzing Fgf ligand genes which are clearly downregulated in *klf2* mutants, we observed a reduction in *fgf14* expression in *npas4l* mutant hearts (*Figure 6—figure supplement 9C*), suggesting its endocardial enrichment. We also observed an

upregulation of *fgf3* expression in *npas4l* mutant hearts (*Figure 6—figure supplement 9C*), suggesting that it is not expressed in the endocardium. Additional studies will be required to identify and analyze the precise role of the *fgf* receptor and ligand genes that potentially function downstream of Klf2 to maintain myocardial wall integrity. Transcriptomic analysis of *klf2* WT and mutant hearts also indicates that genes associated with adherens junctions and focal adhesions are affected in *klf2* mutants, further suggesting the importance of Klf2 in myocardial wall integrity (*Figure 6—figure supplement 2*). Overall, our data suggest that endocardial *klf2* regulates myocardial wall integrity at least in part by modulating myocardial Fgfr signaling.

## Discussion

KLF2, a well-known molecular transducer of fluid shear stress, is vital for cardiovascular development (*Lee et al., 2006*; *Wu et al., 2008*; *Goddard et al., 2017*). Given the embryonic lethality of mouse *Klf2* mutants (*Lee et al., 2006*; *Wu et al., 2008*), we mutated both *klf2a* and *klf2b* genes in zebrafish. Although *klf2a* knockdown by antisense morpholino has been reported to cause impaired valvulogenesis (*Vermot et al., 2009*), single *klf2a^bns11^* or *klf2b^bns12^* mutants did not exhibit any gross morphological or cardiovascular defects, consistent with previous observations of *klf2a^sh317^* mutants and *klf2b* morphants (*Novodvorsky et al., 2015*). In contrast to these findings, a recent study reported that around 80% of 96 hpf *klf2a^ig4^* mutants exhibit a range of valvular defects, although milder than those in *klf2a* morphants (*Vermot et al., 2009*; *Steed et al., 2016b*). Additional analyses will be required to examine the potential difference between the various *klf2a* mutant alleles. Along these lines, since *klf2a^bns11^* and/or *klf2b^bns12^* might be hypomorphic alleles, we further examined these loci. If the first ATG downstream of the *klf2a bns11* or *klf2b bns12* deletion was used to reinitiate translation, a protein lacking the transactivation domain and part of the transrepression domain would be made, and thus such a protein would be lacking at least some of its critical domains. We also found a significant reduction in *klf2a* mRNA levels in *klf2a^bns11^* mutants compared to WT, indicating mRNA decay. *klf2b* and *klf4a* mRNA levels are increased in *klf2a^bns11^* mutants compared to WT, suggesting that they could be compensating for the loss of Klf2a function (*Rossi et al., 2015*). Altogether, these data suggest that *klf2a^bns11^* and *klf2b^bns12^* are strong alleles, although it might be informative to generate promoter-less alleles to further study the role of these genes (*El-Brolosy et al., 2018*).

Interestingly, we found that *klf2* double mutants exhibit a cardiomyocyte extrusion phenotype towards the abluminal side and that this phenotype was dependent on cardiac contractility. Although cardiomyocytes are not considered classical epithelial cells, they display several epithelial features including morphology, polarity, (*Liu et al., 2010*; *Staudt et al., 2014*; *Jiménez-Amilburu et al., 2016*), and N-cadherin distribution (*Cherian et al., 2016*). Cell extrusion is a crucial process that epithelial layers use to maintain barrier properties (*Marinari et al., 2012*; *Kuipers et al., 2014*). This process can be induced by cell death, crowding or loss of cell adhesion molecules (*Rosenblatt et al., 2001*; *Semenza, 2008*; *Eisenhoffer et al., 2012*; *Marinari et al., 2012*; *Kuipers et al., 2014*). Using high-resolution microscopy, we show that, despite the maintenance of apicobasal polarity, cardiomyocyte extrusion in *klf2* mutants correlates with the mislocalization of N-cadherin from the lateral to the apical side of cardiomyocytes. We also observed cardiomyocyte extrusion in *cdh2* morphants, further suggesting the importance of N-cadherin in myocardial wall integrity. These findings are consistent with studies where N-cadherin-negative cardiomyocytes become rounded and loosely aggregated in zebrafish (*Bagatto et al., 2006*), and extruded from chimeric hearts in mouse (*Kostetskii et al., 2001*). A noticeable reduction in ventricular wall thickness was observed in global and endothelial-specific *Klf2* mouse mutants, albeit no significant changes in cardiomyocyte proliferation or death were reported (*Lee et al., 2006*). Therefore, it will be interesting to test whether similar cardiomyocyte extrusion or mislocalization of N-cadherin also occurs in mouse *Klf2* mutants. KLF2 has also been implicated in maintaining endothelial barrier integrity, as *KLF2* loss- or gain-of-function respectively enhances or prevents endothelial leakage in HUVECs upon thrombin treatment (*Lin et al., 2010*). Thus, it will also be interesting to investigate how Klf2 modulates endocardial cell adhesion and barrier integrity.

Cardiac jelly remodeling has been reported to be involved in the maturation of vertebrate hearts (*Nakamura and Manasek, 1981*; *Ramasubramanian et al., 2013*; *Rasouli and Stainier, 2017*; *Grassini et al., 2018*; *Del Monte-Nieto et al., 2018*). Although cardiac jelly thickness appears

unaffected in *klf2* mutants, our study does not exclude the potential role of ECM components in the appearance of the extruding cardiomyocytes. In addition, we observed that the epicardium is affected in *klf2* mutant hearts. Detachment of the epicardium from the myocardial surface was also reported in *Klf2* mutant hearts in mouse (*Lee et al., 2006*). Therefore, it will be interesting to investigate how the lack of Klf2 function results in impaired epicardial coverage in both species. One possibility is that Klf2 cell-autonomously affects the behavior of epicardial cells, or it could be an indirect effect resulting from the morphological changes of the myocardial wall as the epicardial defect in *klf2* mutants was rescued by endothelial/endocardial *klf2b* overexpression, or a combination of both. New tools are required to test these hypotheses in zebrafish.

Taking cell transplantation and gain-of-function approaches, we found that Klf2 cell non-autonomously maintains the integrity of the myocardial wall as endothelial, but not myocardial, *klf2b* overexpression can rescue the extruding cardiomyocyte phenotype in *klf2* mutants, and allow them to survive to adulthood. We also tried to generate an endothelial specific *klf2a* overexpression transgenic line but were unable to do so as the injected fish did not survive. One possibility is that *klf2a* overexpression in endothelial cells leads to their dysfunction or death.

Transcriptome analysis of *klf2* mutant hearts revealed changes in multiple signaling pathways including RA, Hh and Fgf. While increasing RA signaling or decreasing Hh signaling did not cause cardiomyocyte extrusion, decreasing Fgf signaling did. Notably, Fgfr inhibition also led to the mislocalization of N-cadherin from the lateral to the apical side of cardiomyocytes. Interestingly, it has been reported that inhibition of Fgf signaling in mouse embryos, by mosaic overexpression of *Sprouty2*, causes a cell-shedding phenotype in the hepatic epithelium (*Calmont et al., 2006*). Transcriptome analysis of *klf2* mutant hearts also indicated adherens junction and focal adhesion defects. Fgf signaling can have cardioprotective effects by modulating the phosphorylation of the gap junction protein connexin 43 (*Srisakuldee et al., 2006*), which itself can interact with N-cadherin in cell-cell adhesion (*Wei et al., 2005*). Interestingly, we also found a downregulation in *klf2* mutant hearts of *cdon*, which is highly expressed in early zebrafish hearts based on single-cell sequencing data (*Wagner et al., 2018*) and encodes a cell adhesion molecule implicated in myoblast and neuronal differentiation via binding to N-cadherin (*Lu and Krauss, 2010*); Cdon-mediated N-cadherin localization has also been reported to be crucial in neural crest cell migration (*Powell et al., 2015*). *Cdon* expression is induced by KLF2 in vitro (*Boon, 2008*). Thus, it will be interesting to investigate whether Klf2 regulates N-cadherin localization by modulating *cdon* expression. We further found that *egr1*, a flow responsive transcription factor gene involved in FGF-dependent angiogenesis during neo-vascularization and tumor growth (*Khachigian et al., 1997*; *Khachigian, 2006*), is downregulated in *klf2* mutant hearts. Therefore, it will also be interesting to investigate whether *klf2* modulates Fgf signaling by regulating *egr1* expression. Altogether, our data suggest that both Klf2 and Fgf signaling are required to maintain myocardial wall integrity, and it will be important to further investigate how endothelial Klf2, and Fgf signaling, modulate this process. Since cardiomyocyte extrusion is a complex phenotype, it is likely that additional signaling pathways function downstream of Klf2 to maintain myocardial wall integrity.

In summary, this study reveals a novel function for endothelial Klf2 in regulating cardiomyocyte behavior, which in turn modulates myocardial wall integrity. It will be important to further investigate endocardial-myocardial-epicardial interactions which are vital for cardiac wall maturation.

## Materials and methods

**Key resources table**

| Reagent type (species) or resource | Designation | Source or reference | Identifiers | Additional information |
|---|---|---|---|---|
| Genetic reagent (*Danio rerio*) | *klf2a^bns11* | (*Kwon et al., 2016*) | | |
| Genetic reagent (*Danio rerio*) | *klf2b^bns12* | (*Kwon et al., 2016*) | | |

*Continued on next page*

*Continued*

| Reagent type (species) or resource | Designation | Source or reference | Identifiers | Additional information |
|---|---|---|---|---|
| Genetic reagent (*Danio rerio*) | Tg(myl7:LIFEACT-GFP)[s974] | (*Reischauer et al., 2014*) | ZFIN ID: ZDB-ALT-150203–1 | |
| Genetic reagent (*Danio rerio*) | Tg(myl7:mCherry-CAAX)[bns7] | (*Uribe et al., 2018*) | ZFIN ID: ZDB-ALT-181102–5 | |
| Genetic reagent (*Danio rerio*) | Tg(myl7-MKATE-CAAX)[sd11] | (*Lin et al., 2012*) | ZFIN ID: ZDB-ALT-120320–1 | |
| Genetic reagent (*Danio rerio*) | Tg(myl7:EGFP-Hsa.HRAS)[s883] | (*D'Amico et al., 2007*) | ZFIN ID: ZDB-ALT-070309–1 | |
| Genetic reagent (*Danio rerio*) | TgBAC(etv2-EGFP)[ci1] | (*Proulx et al., 2010*) | ZFIN ID: ZDB-ALT-110131–53 | |
| Genetic reagent (*Danio rerio*) | Tg(−0.8myl7:nlsDsRed Express)[hsc4] | (*Takeuchi et al., 2011*) | ZFIN ID: ZDB-ALT-110222–3 | |
| Genetic reagent (*Danio rerio*) | Tg(myl7:mVenus-gmnn)[ncv43] | (*Jiménez-Amilburu et al., 2016*) | ZFIN ID: ZDB-ALT-170131–3 | |
| Genetic reagent (*Danio rerio*) | TgBAC(cdh2:cdh2-EGFP,crybb1:ECFP)[zf517] | (*Revenu et al., 2014*) | ZFIN ID: ZDB-ALT-141218–5 | |
| Genetic reagent (*Danio rerio*) | Tg(hsp70:dn-fgfr1-EGFP)[pd1] | (*Lee et al., 2005*) | | |
| Genetic reagent (*Danio rerio*) | Tg(fli1a:klf2b-p2a-td Tomato)[bns235] | This paper | | Details in Materials and methods |
| Genetic reagent (*Danio rerio*) | Tg(myl7:klf2a-p2a-td Tomato)[bns200] | This paper | | Details in Materials and methods |
| Genetic reagent (*Danio rerio*) | Tg(myl7:klf2b-p2a-td Tomato)[bns234] | This paper | | Details in Materials and methods |
| Genetic reagent (*Danio rerio*) | Tg(−0.2myl7:tdTomato-podxl)[bns197] | This paper | | Details in Materials and methods |
| Genetic reagent (*Danio rerio*) | TgBAC(tcf21:NLS-EGFP)[pd41] | (*Wang et al., 2015*) | ZFIN ID: ZDB-ALT-110914–2 | |
| Genetic reagent (*Danio rerio*) | TgBAC(tcf21:mCherry-NTR)[pd108] | (*Wang et al., 2015*) | ZFIN ID: ZDB-ALT-150904–1 | |
| Sequence-based reagent | tnnt2a morpholino | (*Sehnert et al., 2002*) | ZFIN ID: ZDB-MRPHLNO-060317–4 | |
| Sequence-based reagent | amhc morpholino | (*Berdougo et al., 2003*) | ZFIN ID: ZDB-MRPHLNO-061110–1 | |
| Sequence-based reagent | cdh2 morpholino | (*Lele et al., 2002*) | ZFIN ID: ZDB-MRPHLNO-060815–1 | |
| Commercial assay or kit | RNA Clean and Concentrator kit | Zymo Research | | |
| Software | ZEN Blue 2012 | Zeiss, Germany | | |
| Software | ZEN Black 2012 | Zeiss, Germany | | |

## Zebrafish lines and constructs

Zebrafish were raised and maintained under standard conditions (*Westerfield, 2000*). All animal experiments were done according to German Animal Protection Laws approved by the local governmental animal protection committee.

## Zebrafish mutant and transgenic lines

We used the following mutant and transgenic lines: *klf2a*[bns11] and *klf2b*[bns12] (*Kwon et al., 2016*), *Tg (myl7:LIFEACT-GFP)*[s974] (*Reischauer et al., 2014*), *Tg(myl7:mCherry-CAAX)*[bns7] (*Uribe et al., 2018*), *Tg(myl7-MKATE-CAAX)*[sd11] (*Lin et al., 2012*), *Tg(myl7:EGFP-Hsa.HRAS)*[s883] (*D'Amico et al., 2007*), *TgBAC(etv2-EGFP)*[ci1] (*Proulx et al., 2010*), *Tg(−0.8myl7:nlsDsRedExpress)*[hsc4] (*Takeuchi et al., 2011*), *Tg(myl7:mVenus-gmnn)*[ncv43] (*Jiménez-Amilburu et al., 2016*), *TgBAC(cdh2:cdh2-EGFP, crybb1:ECFP)*[zf517] (*Revenu et al., 2014*), *Tg(hsp70:dn-fgfr1-EGFP)*[pd1] (*Lee et al., 2005*), *Tg(fli1a: klf2b-p2a-tdTomato)*[bns235], *Tg(myl7:klf2a-p2a-tdTomato)*[bns200], *Tg(myl7:klf2b-p2a-tdTomato)*[bns234], *Tg(−0.2myl7:tdTomato-podxl)*[bns197], *TgBAC(tcf21:NLS-EGFP)*[pd41] and *TgBAC(tcf21:mCherry-NTR)* [pd108] (*Wang et al., 2015*).

Generation of *klf2a*[bns11] and *klf2b*[bns12] mutants *klf2a*[bns11] and *klf2b*[bns12] mutants were generated by TALEN mutagenesis targeting exon 2. The following TALEN arms were constructed and assembled using the Golden Gate method (*Cermak et al., 2011*).

For *klf2a*[bns11]:

TALEN arm1: NN NI HD NI HD HD NG NI HD NG NN HD HD NI NI HD HD NN NG HD NG

TALEN arm2: NN NN NN NN NI NI NI NN HD NI NN NN HD HD NG NN NI HD NG

For *klf2b*[bns12]:

TALEN arm1: NN NN HD NI HD NG NN NI NI HD NI HD NI NN NI HD NI HD

TALEN arm2: NG NN HD NG NN NI NN NI NG HD HD NG HD NN NG HD NI NG HD HD NG

100 pg of total TALEN RNA and 50 pg of GFP RNA (used to monitor injection efficiency) were co-injected into the cell at the one-cell stage. The *klf2a*[bns11] and *klf2b*[bns12] alleles were genotyped using high-resolution melt analysis (HRMA) with an Eco Real-Time PCR System (Illumina). The HRMA primers are listed in *Supplementary file 1*.

## Generation of transgenic lines

To generate a myocardial specific *klf2a* or *klf2b* overexpressing construct, the coding sequence of *klf2a* or *klf2b* fused with a self-cleaving peptide 2A and tdTomato, was cloned in a mini tol2 plasmid harbouring a *myl7* promoter using the Cold Fusion Cloning Kit (System Bioscience; MC101A-1-SB). We used the same cloning approach to generate an endothelial-specific *klf2b* overexpression line by putting *klf2b-p2a-tdTomato* under control of a *fli1a* promoter.

To generate the *Tg(−0.2myl7:tdTomato-podxl)* line, we constructed *−0.2myl7:tdTomato-podxl* by tagging the signal peptide of zebrafish *podocalyxin* with *tdTomato* and then cloned it into a Tol2 enabled vector with the *−0.2myl7* promoter using the Cold Fusion Cloning Kit.

A total of 15 pg of each aforementioned DNA construct was co-injected with 10 ng of Tol2 RNA into the cell at the one-cell stage. To establish the *Tg(fli1a:klf2b-p2a-tdTomato)*[bns235], *Tg(myl7:klf2a-p2a-tdTomato)*[bns200], *Tg(myl7:klf2b-p2a-tdTomato)*[bns234], and *Tg(−0.2myl7:tdTomato-podxl)*[bns197] lines, injected larvae (F0) were screened for tdTomato fluorescence and raised to adulthood.

## Quantitative PCR analysis

qPCR for cardiac expression of *fgf* ligand and receptor genes was performed in a CFX Connect Real-Time System (Biorad). RNA was isolated using TRIzol (Life Technologies) followed by phenol-chloroform extraction. In brief, 25 75 hpf hearts were lysed and homogenized in TRIzol using the NextAdvance Bullet Blender Homogenizer (Scientific instrument services). Chloroform was then added and phase separation was obtained following vortexing and centrifugation. The top (aqueous) phase (containing RNA) was isolated and purified using a Clean and Concentrator kit (Zymo). Purified RNA was then used for reverse transcription using the Maxima First Strand cDNA synthesis kit (Thermo). All reactions were performed in at least technical duplicates and the results represent biological triplicates. Gene expression values were normalized using the zebrafish *rpl13* gene. The qPCR primers are listed in *Supplementary file 1*. Ct and dCt values are listed in *Supplementary file 3*. qPCR analysis for global expression of *klf2a*, *klf2b*, and *klf4a* was performed on cDNA obtained

from 24 hpf *klf2a* WT and mutants and 72 hpf *klf2b* WT and mutants. RNA was extracted using a RNeasy Micro kit (QIAGEN). Gene expression values were normalized using the zebrafish *18S ribosomal RNA* and *gapdh* genes. All reactions were performed in three technical replicates, and the results represent three independent biological samples (25 embryos were pooled for each sample). The qPCR primers are listed in *Supplementary file 1*. Ct and dCt values are listed in *Supplementary file 3*.

## Morpholinos

1 nl in total volume of the following morpholinos (MOs) were injected at the one-cell stage: *tnnt2a* MO - 0.5 ng (5'-CATGTTTGCTCTGATCTGACACGCA-3') (*Sehnert et al., 2002*) *amhc* MO - 1 ng (5'-ACTCTGCCATTAAAGCATCACCCAT-3') (*Berdougo et al., 2003*) *cdh2* MO - 1.5 ng (5'-TCTGTATAAAGAAACCGATAGAGTT-3') (*Lele et al., 2002*).

## Chemical treatments

To prevent contractility for 2 hr, 100 hpf larvae were treated with 20 mM BDM (*Higuchi and Takemori, 1989*). To block Fgf and Hh signalling, zebrafish larvae were respectively exposed to SU5402 (15 µM) and Cyclopamine (5 and 10 µM) (Sigma-Aldrich) from 75 to 96 hpf. To increase retinoic acid signaling, dechorionated embryos were treated with 0.5 and 0.75 µM all-trans-retinoic acid (Sigma-Aldrich) from 74 to 96 hpf.

## Heat-shock treatments

For heat-shock treatments, 75 hpf larvae were heat-shocked at 39°C for 1 hr. After treatment, the larvae were incubated at 28°C and imaged at 96 hpf.

## Acridine orange staining

96 hpf larvae were stained with acridine orange (Sigma) by placing them in a solution of the dye in E3 medium (2 mg/mL) for 30 min. The larvae were then washed carefully with E3 medium three times and imaged using confocal microscopy.

## Heart isolation, RNA extraction and microarray profiling

500 hearts from 96 hpf *klf2* WT and mutant larvae were extracted, pooled in 1.5 mL and frozen in −80°C as described (*Burns and MacRae, 2006*). RNA was isolated using Qiazol solution (Qiagen), followed by phenol-chloroform extraction. Briefly, pooled hearts from 96 hpf larvae were first mechanically dissociated and homogenized in Qiazol using the NextAdvance Bullet Blender Homogenizer (Scientific instrument services, inc.). Chloroform was then added and phase separation was obtained following vortexing and centrifugation. The top (aqueous) phase (containing RNA) was isolated and purified using a miRNeasy micro kit (Qiagen).

Microarray analysis was performed by Oak Labs (Germany). The Low Input QuickAmp Labeling Kit (Agilent Technologies) was used to generate fluorescent cRNA (complementary RNA) following manufacturer's protocol. For 1 st strand synthesis, oligo-dT primer or a random primer/oligo-dT primer mixture (WT primer) was used. After 2nd strand synthesis, an in vitro transcription for synthesis of cRNA labelled with cyanine 3-CTP was performed. The Agilent Gene Expression Hybridisation Kit (Agilent Technologies) was used following manufacturer's protocol. cRNA was hybridised on a 8 × 60K microarray at 65°C for 17 hr using Agilent's recommended hybridisation chamber and oven. Finally, the microarray was washed once with the Agilent Gene Expression Wash Buffer one for one minute at ambient temperature followed by a second wash with preheated (37°C) Gene Expression Wash Buffer two for 1 min. Fluorescence signals on microarrays were detected by the SureScan Microarray Scanner G2600D (Agilent Technologies) at a resolution of 3 micron for SurePrint G3 Gene Expression Microarrays, generating a 20 bit TIFF file. The scanned images were analyzed with Feature Extraction Software 11.5.1.1 (Agilent) using default parameters to obtain background subtracted and spatially detrended processed signal intensities. The processed signal values were quantile normalized. In order to allow analyses at the gene level, the normalized probe values were collapsed to the isoform showing the highest mean signal considering all samples per gene.

## Microarray gene set enrichment analysis (GSEA)

The Broad GSEA tool version 3 (*Subramanian et al., 2005*) was used to identify up- and down-regulated gene sets. The analysis was based on a pre-ranked list of genes scored by the log2 fold change multiplied with the mean normalized signal value. 1000 permutations were performed with a maximum gene set size of 500 and classic weighting versus hallmark and curated gene sets. Downregulated pathways in *klf2* mutants are listed in *Supplementary file 2*.

## Whole mount in situ hybridization

Whole mount in situ hybridization was performed as described before (*Thisse and Thisse, 2008*). 36, 48 and 72 hpf animals were fixed in 4% paraformaldehyde overnight at 4°C and subsequently dehydrated in 100% methanol at −20°C. Samples were rehydrated with 1X phosphate buffered saline (PBS) and permeabilized with proteinase K (10 μg/mL) at RT for 30 min. After washing with 1X PBT (1X PBS, 0.1% Tween 20 (vol/vol)), larvae were hybridized with 150 ng of *klf2b* antisense DIG-labeled RNA probe overnight at 70°C. The hybridized probes were then detected with alkaline phosphatase conjugated anti-Digoxigenin antibody (Roche, dilution 1:1,000) for 3 hr at RT, and the signal was visualized with BM purple (Roche). The probe for *klf2b* was amplified from 48 hpf heart cDNA using *klf2b* forward 5'-TGGCTTTACCTTGCCTTTTG −3' and reverse 5'- CCGTGTGTTTACGGAAGTGA −3' primers. The PCR fragment was then subcloned into pGEM-T.

## Larval heart immunostaining

Right after stopping heart contraction with 0.2 percent tricaine, larvae were fixed in fish fix (4% paraformaldehyde (PFA)) overnight at 4°C. Fixed larvae were then washed three times with PBS/0.1% Tween. Washed larvae were then deyolked with forceps and permeablized with Proteinase K solution (3 μg/mL) in 1 mL PBS/Tween for 3 hr at RT. Further, the larvae were washed with PBDT (PBS/1% BSA/1% DMSO/0.5% Triton-X 100) and incubated in 2 mL of blocking solution (1.62 mL PBS/0.1% Tween +200 uL sheep serum +80 uL 20% Triton-X +100 μL 20% BSA) for 3 hr at RT. Next, the larvae were incubated in primary antibody overnight at 4°C. After washing with PBDT (four times), the larvae were incubated in secondary antibody for 3 hr at RT. For DAPI staining, larvae were incubated with DAPI (0.01 mg/mL in PBS, Sigma) and washed carefully with PBS/Tween (five times). After washing, larvae were mounted in 2% low melting agarose for imaging. Primary antibodies used for immunofluorescence were anti-GFP at 1:500 (Chicken; Aves Labs), anti-mCherry at 1:100 (mouse; Clontech), and anti-pERK (p44/42 MAPK (Erk1/2)) at 1:100 (rabbit, Cell Signaling). Secondary antibodies were used at 1:500 (Life Technologies).

## Adult heart histology and immunostaining

This protocol was adapted from *Mateus et al. (2015)* with the following modification: after 1 hr fixation with 4% PFA, adult hearts were saturated overnight in 30% sucrose (in PBS). Embedding was done in 7.5% gelatin in 15% sucrose (in PBS), followed by freezing in isopentane cooled in liquid nitrogen. Using a Leica Microtome, the hearts were sectioned at 10 μm and maintained at −20°C until further use. Prior to immunostaining, sections were thawed for 10 min at room temperature and washed twice in PBS at 37°C for 10 min to remove gelatin. Subsequently, sections were washed once in 0.1M glycine for 10 min and permeabilized in acetone for 7 min at −20°C. Sections were washed in 1% Bovine Serum Albumin (BSA, Sigma), 1% DMSO (Sigma) and 0.1% Triton-X100 in 0.5xPBS (PBDX), followed by 2 hr of blocking in PBDX with 5% goat serum. The samples were incubated overnight with phalloidin (1:200, conjugated with Alexa Fluor 488, Invitrogen) and anti-Caveolin one antibody (BD Biosciences, 1:200) at 4°C and counterstained with DAPI (0.001 mg/mL in PBS, Sigma). Secondary antibodies were used at 1:500 (Life Technologies). Slides were mounted with DAKO Fluorescence Mounting Media and imaged using a Zeiss LSM800 confocal microscope.

For Hematoxylin and Eosin staining (H and E), cryosections were stained for 10 min with acidic hemalum (Wladeck), washed in running tap water until sections turned blue and rinsed in deionized water. Sections were subsequently stained for 6 min with an eosin solution (Waldeck), dehydrated in ethanol and cleared in roti-histol prior to mounting in Entellan (Merck). Sections were imaged using a Nikon SMZ25 and the NIS Elements software, and processed using FIJI.

## Imaging, quantification and data processing

Zebrafish embryos and larvae were mounted and anesthetized in 2% low-melt agarose (Sigma) containing 0.2% tricaine on glass-bottom dishes. While mounting, samples were manually oriented towards the microscope lens to enhance optical access to the heart. Images were captured with Zeiss LSM780, LSM800, LSM 880 and spinning disk (CSU-X1 Yokogawa) confocal microscopes at 40X magnification. The confocal data were then processed with the ZEN 2012 software (black and blue edition). Adult fish and dissected hearts were imaged using a Nikon SMZ25 and the NIS Elements software and processed using FIJI software.

## Cardiomyocyte size and circularity

Size and circularity of compact layer cardiomyocytes in the outer curvature of the ventricle were measured with ImageJ.

## Cell transplantation

WT embryos were obtained from *Tg(myl7:MKATE-CAAX)* incrosses. Mutant and heterozygous embryos were obtained by incrossing *Tg(myl7:LIFEACT-GFP); klf2⁺/⁻* animals. All embryos were dechorionated in pronase (1 mg/mL) for 5 min at 28°C and then incubated in agarose-coated dishes with 1/3 Ringer solution supplemented with penicillin (50 U/mL)/streptomycin (50 µg/mL). Donor cells were taken from embryos at mid-blastula stages and transplanted into age-matched host embryos. Using confocal microscopy, the chimeric hearts were then imaged at 96 hpf. Donors or hosts obtained from incrossing *Tg(myl7:LIFEACT-GFP); klf2⁺/⁻* animals were used for genotyping.

## Statistical analysis

Data were analyzed with the Prism six software. Values are presented as mean ±s.e. *P values* (*$p \leq 0.05$, **$p \leq 0.01$, ***$p \leq 0.001$) were calculated by two-tailed Student's *t* test.

## Acknowledgements

We acknowledge Albert Wang for critical reading of the manuscript, Radhan Ramadass for confocal imaging and technical support, Oliver Stone, Vanesa Jimenez-Amilburu, Jason Lai, Sri Teja Mulla-pudi, Michele Marass, Sven Reischauer, Arica Beisaw and Carol Yang for sharing reagents and helpful discussions, Sharon Meaney-Gardian and Simon Perathoner for excellent assistance, as well as Jean-Francois Matuszek, Rita Retzloff and Martin Laszczyk for zebrafish care. PG thanks Thomas Braun for support. We would also like to thank the reviewing editor and the reviewers for their constructive feedback. This work was supported in part by the Max Planck Society and the Leducq Foundation. SJR is a graduate student registered with the Faculty of Biological Sciences at Goethe University, Frankfurt am Main.

## Additional information

### Competing interests

Didier Y Stainier: Senior editor, *eLife*. The other authors declare that no competing interests exist.

### Funding

| Funder | Grant reference number | Author |
|---|---|---|
| Fondation Leducq | | Didier Y Stainier |
| Max-Planck-Gesellschaft | Open-access funding | Didier Y Stainier |

The funders had no role in study design, data collection and interpretation, or the decision to submit the work for publication.

## Author contributions

Seyed Javad Rasouli, Conceptualization, Data curation, Formal analysis, Validation, Investigation, Visualization, Methodology, Writing—original draft, Writing—review and editing, Performed heart extraction, larval and adult immunostaining, Performed whole mount in situ hybridization and qPCR experiments, Analysed microarray data and performed the rest of the experiments; Mohamed El-Brolosy, Conceptualization, Formal analysis, Validation, Methodology, Writing—review and editing, Performed whole mount in situ hybridization and qPCR experiments; Ayele Taddese Tsedeke, Conceptualization, Methodology, Writing—review and editing, Perfomed heart extraction; Anabela Bensimon-Brito, Conceptualization, Formal analysis, Investigation, Methodology, Writing—review and editing, Performed adult immunostaining; Parisa Ghanbari, Conceptualization, Formal analysis, Investigation, Methodology, Writing—review and editing, Performed heart extraction and larval immunostaining; Hans-Martin Maischein, Conceptualization, Investigation, Performed the transplantation experiments; Carsten Kuenne, Conceptualization, Software, Formal analysis, Validation, Writing—review and editing, Analyzed microarray data; Didier Y Stainier, Conceptualization, Supervision, Funding acquisition, Validation, Writing—original draft, Writing—review and editing

## Author ORCIDs

Seyed Javad Rasouli (iD) http://orcid.org/0000-0002-0225-2565
Ayele Taddese Tsedeke (iD) https://orcid.org/0000-0001-7493-2511
Anabela Bensimon-Brito (iD) http://orcid.org/0000-0003-1663-2232
Parisa Ghanbari (iD) http://orcid.org/0000-0002-7432-2147
Didier Y Stainier (iD) http://orcid.org/0000-0002-0382-0026

## Ethics

Animal experimentation: All animal experiments were done in accordance with institutional (MPG) and national ethical and animal welfare guidelines approved by the ethics committee for animal experiments at the Regierungspräsidium Darmstadt, Germany (permit numbers B2/1017, B2/1041, B2/1138 and B2/Anz. 1007).

## Decision letter and Author response

Decision letter https://doi.org/10.7554/eLife.38889.040
Author response https://doi.org/10.7554/eLife.38889.041

# Additional files

## Supplementary files

• Supplementary file 1. List of qPCR primers.
DOI: https://doi.org/10.7554/eLife.38889.033

• Supplementary file 2. Downregulated pathways in *klf2* mutants.
DOI: https://doi.org/10.7554/eLife.38889.034

• Supplementary file 3. Ct and dCt values of qPCR data.
DOI: https://doi.org/10.7554/eLife.38889.035

• Transparent reporting form
DOI: https://doi.org/10.7554/eLife.38889.036

## Data availability

The authors declare that all data supporting the findings of this study are available within the article and its supplementary information files. Microarray data have been deposited in GEO under accession number GSE122137.

The following dataset was generated:

| Author(s) | Year | Dataset title | Dataset URL | Database and Identifier |
|---|---|---|---|---|
| Rasouli SJ, El-Brolosy M, Taddese | 2018 | Microarray data from The flow responsive transcription factor Klf2 | https://www.ncbi.nlm.nih.gov/geo/query/acc. | NCBI Gene Expression Omnibus, |

| Tsedeke A, Bensi-mon-Brito A | is required for myocardial wall integrity by modulating Fgf signaling | cgi?acc=GSE122137 | GSE122137 |

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
