## [Decision Letter]

Thank you for submitting your manuscript "The flow responsive transcription factor Klf2 is required for myocardial wall integrity by modulating FGF signaling" for consideration for publication in *eLife*. Your article has been reviewed by three peer reviewers, and the evaluation has been overseen by Deborah Yelon as the Reviewing Editor and Anna Akhmanova as the Senior Editor. The following individual involved in review of your submission has agreed to reveal their identity: Mingfu Wu (Reviewer #2).

The reviewers have discussed their reviews with one another and the Reviewing Editor has drafted this decision to help you with the preparation of a revised submission. The reviewers and editor have identified a number of additional tasks that we hope you can achieve in a reasonable length of time.

Summary:

In this manuscript, Rasouli and colleagues examine the role of the flow-responsive *klf2* genes during zebrafish cardiac development. They find that *klf2a* and *klf2b* are genetically redundant, and that simultaneous loss-of-function of both genes leads to variable cardiac defects, most notably the extrusion of ventricular cardiomyocytes "abluminally" around 4 days post fertilization. This abluminal extrusion requires cardiac contractility and is regulated by *klf2* genes in a non-autonomous fashion. In addition, cardiomyocyte extrusion is accompanied by the abnormal organization of N-cadherin, suggesting that adherens junctions are key to mediating the myocardial integrity that is disrupted in double mutants. Endocardial overexpression (but not myocardial overexpression) of *klf2b* rescues the extrusion phenotype, indicating that endocardial expression of *klf2* is important for avoiding ventricular disruption. Finally, the authors argue that *klf2* is acting through FGF/Erk signaling to regulate myocardial integrity: they find that inhibition of FGF signaling can genetically interact with reduced *klf2* levels in a dose-dependent manner, inducing cell extrusion, and that inhibition of FGF alone can also recapitulate the phenotype and affect N-cadherin localization. Altogether, the manuscript is well-written, the imaging is high-quality, and the take-home message is compelling. The authors' data tell an intriguing and meaningful story about the mechanisms regulating tissue integrity in the heart during myocardial wall maturation, as well as the importance of endocardial-myocardial communication and the relationship between cardiac function and morphology during this process. However, some elements of the mechanism underlying the double mutant phenotype could benefit from further clarification and articulation in order to strengthen the significance and value of this work for the readers of *eLife*.

Essential revisions:

1) Additional quantification of the phenotype would be helpful, particularly in terms of quantifying how the phenotype emerges over time, as this would clarify the specific onset of the phenotype and how to evaluate the degree of phenocopy in various scenarios. For example, the authors state that "the number of extruding cardiomyocytes […] is variable and increases over time." Could this be more clearly quantified? If it increases over time, by how much and over what time-scale? What is the level of variation? How does this compare with the quantification and timing of cell extrusion when FGF signaling is inhibited? It appears as though the cell adhesion/morphology phenotype is more severe in the *klf2a/b* double mutants than in the FGF inhibition scenario. Is this true? If so, FGF is unlikely to be the only downstream factor affecting cell extrusion, and discussion of this point would be appropriate.

2) While the authors focus on the cellular features of the extruding cells, the general population of double mutant ventricular cardiomyocytes appears smaller and more circular than in wild-type. Has this aspect of the phenotype been quantified? Might there be a broader myocardial phenotype, and, if so, when does this phenotype emerge? Since the authors state that extrusion increases over time, it seems possible that the abnormally shaped ventricular cardiomyocytes could represent an intermediate population on their way to extrusion.

3) In evaluating whether defects in the cardiac jelly might contribute to the cardiomyocyte extrusion phenotype, the authors focus on analyzing the cardiac jelly thickness. Of course, other aspects of the ECM could be affected here, such as expression/distribution of specific components (as mentioned in the Discussion) or stiffness. Considering that *klf2a* has been suggested to regulate fibronectin expression and that the ECM is a likely mediator of endocardial-myocardial communication, additional information about the integrity of the ECM in double mutants would be appreciated. Could the authors use immunostaining to examine ECM components such as fibronectin or versican?

4) It would be helpful to have more information about how fully the *klf2* double mutant phenotype is rescued by endocardial expression of *klf2b*. For example, the authors report an apparent decrease in ventricular proliferation in double mutants, but dismiss it as a cause of the extrusion phenotype (arguing that only an increase in proliferation would lead to extrusion). Is this aspect of the phenotype rescued with overexpression of *klf2b* in endocardial cells? Also (and importantly), are the epicardial defects observed in double mutants rescued by endocardial overexpression of *klf2b*? These data may help to distinguish between different possibilities regarding the role of *klf2* genes during epicardial development. Connected to this, it would be valuable to better understand whether the incomplete epicardial coverage seen in *klf2* double mutants could contribute to their myocardial extrusion phenotype. Do cardiomyocytes extrude in other mutants with epicardial coverage defects?

5) The authors observe reduction of pERK in *klf2* double mutants and find that pERK levels are rescued by *klf2* overexpression. However, ERK is downstream of multiple pathways. Does expression of *dnfgfr* also reduce the pERK signal in this context?

6) It would be valuable for the text to more clearly articulate the authors' proposed model (or models) for the relationship between *klf2* genes, FGF signaling, and myocardial integrity. For example, is it that *klf2* induces FGF ligand transcription in the endocardium, which signals in an autocrine fashion and is transduced by pErk, and that this endocardial response is then somehow transmitted to the myocardium to sustain its integrity? Or, is the model that a myocardial FGF ligand(s) is somehow induced by endocardial *klf2*, and then signals back to the endocardium to signal through Fgfr/Erk? Would expression analysis of the downregulated FGF ligands in transverse sections of the ventricle help to distinguish between possible models Also, these models would presumably involve an additional signal that crosses the cardiac jelly to signal to the myocardium (and potentially the epicardium), downstream of *klf2*>*FGF*. Could the authors discuss what this signal might be? Finally, how do the authors envision that the dynamic localization of N-Cadherin affects cell behavior? In the mouse heart, N-Cadherin has been suggested to be involved in establishing polarity and promoting cell migration. Are these likely roles of N-Cadherin during myocardial wall maturation in zebrafish?

---

## [Author Response]

Essential revisions:1) Additional quantification of the phenotype would be helpful, particularly in terms of quantifying how the phenotype emerges over time, as this would clarify the specific onset of the phenotype and how to evaluate the degree of phenocopy in various scenarios. For example, the authors state that "the number of extruding cardiomyocytes […] is variable and increases over time." Could this be more clearly quantified? If it increases over time, by how much and over what time-scale? What is the level of variation? How does this compare with the quantification and timing of cell extrusion when FGF signaling is inhibited? It appears as though the cell adhesion/morphology phenotype is more severe in the klf2a/b double mutants than in the FGF inhibition scenario. Is this true? If so, FGF is unlikely to be the only downstream factor affecting cell extrusion, and discussion of this point would be appropriate.

Additional quantification of the cardiomyocyte extrusion phenotype in *klf2* mutants and FGFR inhibited animals has been done, and these new data are shown in Figure 1—figure supplement 4 and Figure 6—figure supplement 6.

In order to further compare the myocardial phenotype in *klf2* mutants and FGFR inhibitor treated animals, we measured the size and circularity of their cardiomyocytes and did not observe any significant differences. These new data are shown in Figure 1—figure supplement 4 and Figure 6—figure supplement 6. In addition, we found that Cdh2-GFP mislocalization occurred in most FGFR inhibitor treated hearts (10/11), and have incorporated this point in the manuscript.

2) While the authors focus on the cellular features of the extruding cells, the general population of double mutant ventricular cardiomyocytes appears smaller and more circular than in wild-type. Has this aspect of the phenotype been quantified? Might there be a broader myocardial phenotype, and, if so, when does this phenotype emerge? Since the authors state that extrusion increases over time, it seems possible that the abnormally shaped ventricular cardiomyocytes could represent an intermediate population on their way to extrusion.

We have quantified the size and circularity of cardiomyocytes in *klf2* WT and mutant animals and these new data are shown in Figure 1—figure supplement 4.

3) In evaluating whether defects in the cardiac jelly might contribute to the cardiomyocyte extrusion phenotype, the authors focus on analyzing the cardiac jelly thickness. Of course, other aspects of the ECM could be affected here, such as expression/distribution of specific components (as mentioned in the Discussion) or stiffness. Considering that klf2a has been suggested to regulate fibronectin expression and that the ECM is a likely mediator of endocardial-myocardial communication, additional information about the integrity of the ECM in double mutants would be appreciated. Could the authors use immunostaining to examine ECM components such as fibronectin or versican?

We fully agree with the reviewers regarding the potential roles of ECM components in endocardial-myocardial communication and cardiomyocyte behavior.

According to our transcriptomic data, the cardiac expression of *fibronectin 1a (fn1a*) and *fibronectin 1b (fn1b*) appear unchanged between 96 hpf *klf2* WT and mutant animals. However, expression of *versican a (vcana*) and *versican b (vcanb*) appear to be up- and down-regulated, respectively (Author response table 1).

**Author response table 1. resptable1:** 

Transcript ID	Gene	*klf2* WT	*klf2* Mut	*klf2* Mut/WT ratio
ENSDART00000124346	*fn1a*	3717	4062	1,09
ENSDART00000010521	*fn1a*	3905	4104	1,05
ENSDART00000023692	*fn1b*	2666	2493	0,93
ENSDART00000103755	*fn1b*	2689	2393	0,88
ENSDART00000099132	*vcana*	452	1053	2,32
ENSDART00000144973	*vcana*	906	2080	2,29
ENSDART00000134304	*vcanb*	401	343	0,85
ENSDART00000012522	*vcanb*	342	210	0,61

Author response table 1. Cardiac expression of *fibronectin* and *versican* in *klf2* WT and mutant hearts.

We also immunostained *klf2* WT and mutant hearts for Fibronectin and Versican and did not detect any obvious differences (data not shown).

4) It would be helpful to have more information about how fully the klf2 double mutant phenotype is rescued by endocardial expression of klf2b. For example, the authors report an apparent decrease in ventricular proliferation in double mutants, but dismiss it as a cause of the extrusion phenotype (arguing that only an increase in proliferation would lead to extrusion). Is this aspect of the phenotype rescued with overexpression of klf2b in endocardial cells? Also (and importantly), are the epicardial defects observed in double mutants rescued by endocardial overexpression of klf2b? These data may help to distinguish between different possibilities regarding the role of klf2 genes during epicardial development. Connected to this, it would be valuable to better understand whether the incomplete epicardial coverage seen in klf2 double mutants could contribute to their myocardial extrusion phenotype. Do cardiomyocytes extrude in other mutants with epicardial coverage defects?

Regarding the proliferation issue, we apologize if the text was not clear. Two studies from our lab, and others, have shown that inhibition of cardiomyocyte proliferation in zebrafish embryos reduces the total number of cardiomyocytes while not leading to their extrusion (Liu et al., 2010; Choi et al., 2013; Lu et al., 2017; Uribe et al., 2018), indicating that a reduction in cardiomyocyte proliferation does not cause their extrusion. We have clarified this point in the manuscript.

We also quantified the number of proliferative cardiomyocytes in *klf2* WT and mutant animals overexpressing *klf2b* in their endothelium and found that the reduction of ventricular cardiomyocyte proliferation observed in *klf2* mutants was rescued by endothelial overexpression of *klf2b*. These new data are shown in Figure 5—figure supplement 2.

Interestingly, endothelial overexpression of *klf2b* also rescued the epicardial defects observed in *klf2* mutants (Figure 5—figure supplement 3); these new data are shown in Figure 5—figure supplement 3. We also stained the epicardium with a Caveolin antibody and found that epicardial coverage was complete in *klf2* mutant adults overexpressing *klf2b* in their endothelium (Figure 5Q-R’).

Overall, these data indicate that endothelial/endocardial *klf2* is required for myocardial wall integrity by regulating endocardial-myocardial-epicardial interactions.

As the reviewers suggested, we also investigated whether incomplete epicardial coverage could contribute to the cardiomyocyte extrusion phenotype observed in *klf2* mutants, using the NTR/MTZ system genetic ablation approach to remove epicardial cells. We found that epicardial ablation could also induce cardiomyocyte extrusion; however, the number of extruding cardiomyocytes after epicardial ablation was far lower than that observed in *klf2* mutants. These new data are shown in Figure 3—figure supplement 4N-R.

5) The authors observe reduction of pERK in klf2 double mutants and find that pERK levels are rescued by klf2 overexpression. However, ERK is downstream of multiple pathways. Does expression of dnfgfr also reduce the pERK signal in this context?

In order to inhibit FGF signaling, we initially treated WT larvae with the FGFR inhibitor SU5402 and observed reduced pERK immunostaining in their hearts, as expected from previous reports (e.g., Shinya et al., 2001; Calmont et al., 2006). We have now also checked pERK immunostaining after *dnfgfr1* overexpression and observed a reduction in pERK immunostaining although milder than that in FGFR inhibitor treated hearts. The new data are shown in Figure 6—figure supplement 7.

6) It would be valuable for the text to more clearly articulate the authors' proposed model (or models) for the relationship between klf2 genes, FGF signaling, and myocardial integrity. For example, is it that klf2 induces FGF ligand transcription in the endocardium, which signals in an autocrine fashion and is transduced by pErk, and that this endocardial response is then somehow transmitted to the myocardium to sustain its integrity? Or, is the model that a myocardial FGF ligand(s) is somehow induced by endocardial klf2, and then signals back to the endocardium to signal through Fgfr/Erk? Would expression analysis of the downregulated FGF ligands in transverse sections of the ventricle help to distinguish between possible models Also, these models would presumably involve an additional signal that crosses the cardiac jelly to signal to the myocardium (and potentially the epicardium), downstream of klf2>FGF. Could the authors discuss what this signal might be?

Fgf signaling, which is essential for endo-myo-epicardial interactions, could be active in any of these three cardiac layers based on the spatiotemporal expression of *fgf* ligand and receptor genes (Sugi et al., 2003; Lavine et al., 2005; Hotta et al., 2008; Lu et al., 2008; Torlopp et al., 2010; Urness et al., 2011; von Gise and Pu, 2012; Itoh and Ohta, 2013; Kennedy-Lydon & Rosenthal, 2015). Transcriptomic analysis of *klf2* WT and mutant hearts shows that Fgf signaling is affected in *klf2* mutants (Figure 6—figure supplement 2, Figure 6—source data 3 and Supplementary file 2), which was confirmed by qPCR (Figure 6— figure Supplement 4). We also found that inhibition of Fgfr signaling led to cardiomyocyte extrusion (Figure 6 and Figure 6—figure supplement 6). By examining single-cell RNA sequencing data of early zebrafish embryos (Wagner et al., 2018), we found that most *fgf* receptor genes and many *fgf* ligand genes are expressed in the zebrafish heart (Figure 6—figure supplement 8). Because the level of expression of most of these genes is rather low, and possibly challenging to detect by in situ hybridization, we performed qPCR analysis in *cloche/npas4l* mutant hearts which lack all endocardial cells (Stainier et al.et al., 1995; Reischauer et al., 2016). Expression levels of all *fgf* receptorgenes were slightly increased in *npas4l* mutants (Figure 6—figure supplement 9B), suggesting that they are not expressed in the endocardium. Analyzing *fgf* ligand genes which are clearly downregulated in *klf2* mutants, we observed a reduction in *fgf14* expression in *npas4l* mutant hearts (Figure 6—figure supplement 9C), suggesting its endocardial enrichment. We also observed an upregulation of *fgf3* expression in *npas4l* mutant hearts (Figure 6—figure supplement 9C), suggesting that it is not expressed in the endocardium. Admittedly, these experiments are not without significant caveats, and BAC reporter lines will need to be generated for all relevant genes to again additional insight into this question. Overall, our data suggest that endocardial *klf2* regulates myocardial wall integrity at least in part by modulating Fgf signaling. However, cardiomyocyte extrusion is a complex phenotype and we do not exclude the potential role of additional signaling pathways downstream of Klf2 function in maintaining myocardial wall integrity.

Finally, how do the authors envision that the dynamic localization of N-Cadherin affects cell behavior? In the mouse heart, N-Cadherin has been suggested to be involved in establishing polarity and promoting cell migration. Are these likely roles of N-Cadherin during myocardial wall maturation in zebrafish?

N-cadherin is highly enriched in mouse and zebrafish cardiomyocytes (Kostetskii et al., 2001; Bagatto et al., 2006; Cherian et al., 2016). These adhesion molecules become mislocalized from the lateral to the apical side of cardiomyocytes in klf2 mutants, suggesting that cardiomyocyte adhesion is affected (Figure 3 J-M’). Cardiomyocyte extrusion was also observed in n-cadherin morphants (Figure 3—figure supplement 1), further suggesting the importance of N-cadherin in cardiomyocyte behavior. These findings are consistent with previous studies describing how Ncadherin-negative cardiomyocytes become rounded and loosely aggregated in zebrafish (Bagatto et al., 2006), and extruded from chimeric hearts in mouse (Kostetskii et al., 2001). One possibility is that mislocalization of N-cadherin molecules in cardiomyocytes affects their lateral adhesion and in turn makes them unstable in this highly contractile organ.